# Upregulation of MMP3 Promotes Cisplatin Resistance in Ovarian Cancer

**DOI:** 10.3390/ijms26094012

**Published:** 2025-04-24

**Authors:** Mariela Rivera-Serrano, Marienid Flores-Colón, Fatima Valiyeva, Loyda M. Meléndez, Pablo E. Vivas-Mejía

**Affiliations:** 1Department of Biology, University of Puerto Rico-Rio Piedras Campus, San Juan 00925, Puerto Rico; riverase@cshl.edu; 2Department of Biochemistry, University of Puerto Rico—Medical Sciences Campus, San Juan 00936, Puerto Rico; marienid.flores@upr.edu; 3Comprehensive Cancer Center, University of Puerto Rico, San Juan 00936, Puerto Rico; fvaliyeva@cccupr.org; 4Translational Proteomics Center, Research Capacity Core, Center for Collaborative Research in Health Disparities, University of Puerto Rico—Medical Sciences Campus, San Juan 00936, Puerto Rico; loyda.melendez@upr.edu; 5Department of Microbiology and Medical Zoology, University of Puerto Rico—Medical Sciences Campus, San Juan 00936, Puerto Rico

**Keywords:** high-grade serous ovarian cancer, MMP3, cisplatin, MMP3 inhibitors, RNA-seq, immunoprecipitation, mass spectrometry

## Abstract

Most women with ovarian cancer (OC) develop resistance to platinum chemotherapy, posing a significant challenge to treatment. Matrix metalloproteinase 3 (MMP3) is overexpressed in High-Grade Serous Ovarian Cancer (HGSOC) and is associated with poor survival outcomes; however, its role in platinum resistance remains underexplored. We evaluated the baseline and cisplatin-induced MMP3 transcript and protein levels in cisplatin-resistant OC cells, revealing significantly higher MMP3 levels in cisplatin-resistant cells than in cisplatin-sensitive cells. siRNA-mediated MMP3 knockdown in cisplatin-resistant OC cells significantly reduced viability, proliferation, and invasion, and these effects were further enhanced when combined with cisplatin treatment, indicating a possible synergistic impact on reducing cancer cell aggressiveness; however, chemical MMP3 inhibition did not replicate these effects. RNA sequencing of MMP3-siRNA-treated cisplatin-resistant HGSOC cells revealed 415 differentially expressed genes (DEGs) compared to the negative control, with an additional 440 DEGs identified in MMP3-siRNA HGSOC cells treated in combination with cisplatin. These DEGs were enriched in pathways related to cell cycle regulation, apoptosis, metabolism, stress response, and extracellular matrix organization. Co-immunoprecipitation-coupled mass spectroscopy (IP-MS) identified MMP3-interacting proteins that may contribute to cell survival and chemoresistance in cisplatin-resistant OC. While MMP3-siRNA monotherapy did not reduce tumor growth in vivo, its combination with cisplatin significantly inhibited tumor growth in a cisplatin-resistant HGSOC xenograft model. These findings underscore the multifaceted role of MMP3 in cisplatin resistance, suggesting its involvement in critical cellular processes driving chemoresistance and highlighting the challenges associated with direct MMP3 targeting in therapeutic strategies.

## 1. Introduction

Ovarian cancer (OC) is the most lethal gynecological malignancy. An estimated 313,000 new OC cases and 207,252 OC-related deaths occurred worldwide in 2020 [1]. High-grade serous ovarian cancer (HGSOC) accounts for 70–80% of all OC deaths. Most women with HGSOC are diagnosed at a late stage when the tumor has already metastasized. The current standard treatment for patients with advanced OC is a combination of cytoreductive surgery and platinum/taxane-based chemotherapy [2]. The platinum atom of cisplatin binds covalently to the N7 position of purines to form interchain cross-links known as cisplatin-DNA adducts, which cause cellular responses such as replication arrest, transcription inhibition, cell cycle arrest, DNA repair, and apoptosis [3]. However, the majority of women with advanced HGSOC develop platinum resistance, which leads to recurrence and disease progression [2]. The cellular mechanisms of cisplatin resistance include decreased cisplatin accumulation, enhanced detoxification systems, increased enzymatic deactivation, accelerated DNA repair, metabolic rewiring, senescence, chromatin remodeling, and epigenetic changes [4,5]. Additional factors include the inactivation of apoptotic pathways, protective autophagy, tumor heterogeneity, interactions with the tumor microenvironment, and dysregulation of oncogenes, tumor suppressor genes, and noncoding RNAs [6,7,8]. Recent treatment approaches, such as PARP inhibitors, angiogenesis inhibitors, and antibody-drug conjugates, have introduced promising new strategies for treating platinum-resistant ovarian cancer, although their impact on overall survival (OS) is still being assessed [9,10,11]. The limited effectiveness of these therapies and the significant challenges they present highlight the urgent need for continued research and the development of targeted treatment strategies with improved efficacy compared to existing therapeutic options.

The deregulation of matrix metalloproteinases (MMPs), a family of zinc-dependent endopeptidases, contributes to virtually all steps of carcinogenesis [12,13]. Most MMPs are secreted and are capable of cleaving extracellular matrix proteins in response to physiological and pathological environmental cues [14,15]. Humans express 23 different MMPs that play major roles in cell differentiation, proliferation, wound healing, apoptosis, and angiogenesis [16]. Accumulating evidence has shown that, owing to these extracellular matrix-degrading functions, increased levels of MMPs accelerate the metastasis and invasion of cancerous cells by facilitating the escape of cancer cells into the surrounding tissues and blood [12,13]. Increased levels of MMPs have been reported in many cancer types, making them attractive therapeutic targets [12,13]. The concept of targeting MMPs for cancer therapy originated over three decades ago with the development of peptidomimetic and non peptidomimetic MMP inhibitors [17]. Most of these inhibitors bind to the zinc ion in the catalytic domain of MMPs to prevent their activity. Initial clinical trials were unsuccessful, as inhibitors failed due to their lack of selectivity and specificity for one MMP, which led to detrimental side effects [18,19].

MMP3 (matrix metalloproteinase 3) is synthesized as an inactive zymogen and is comprised of three structural domains: the N-terminal propeptide domain, the catalytic domain, and the hemopexin domain. The N-terminal propeptide domain contains a cysteine switch motif that maintains MMP3 in its inactive form by binding to the zinc ion in the catalytic domain, preventing its proteolytic activity. Upon activation, the catalytic domain becomes functional and facilitates the hydrolysis of extracellular matrix components, contributing to tissue remodeling and cell migration. The hemopexin domain, with its large surface area, mediates protein-protein interactions and enables MMP3 to regulate signaling pathways independently of its proteolytic activity. This structural complexity underpins MMP3’s dual functionality in both extracellular matrix degradation and signaling regulation [20]. Given its association with different biological processes and pathologies, MMP3 has been proposed as a potential biomarker and therapeutic target for periodontitis, thrombosis, cardiovascular disease, and colon, ovarian, and lung cancer [21,22,23,24,25,26]. The overexpression of MMP3 and its function as a prognostic factor have been reported in breast, cervical, renal, colorectal, gastric, lung, melanoma, pancreatic, and ovarian carcinomas [27,28,29,30].

Recently, we reported that MMP3 is highly abundant in cisplatin-resistant OC cells compared to cisplatin-sensitive OC cells [31]. To determine the contribution of MMP3 to cisplatin resistance in OC, we considered a more holistic approach to specifically inhibit MMP3 using small interfering RNA (siRNA) molecules.

Small interfering RNA (siRNA) is a widely used RNA interference (RNAi) tool for sequence-specific gene knockdown. Processed from double-stranded RNA by DICER, siRNA guides the RNA-induced silencing complex (RISC) to degrade target mRNAs with sequence complementarity or block their translation via the endonuclease AGO2 [32]. Given the critical need for targeted therapies in OC, siRNA-based approaches have gained traction, with several candidates advancing through clinical trials and some receiving regulatory approval [33].

In this study, siRNA-mediated knockdown of MMP3 was employed to evaluate the impact of targeting MMP3 on cisplatin resistance in OC. We hypothesize that the upregulation of MMP3 and its downstream effectors will contribute to cisplatin resistance in OC. To test this, we studied the biological and functional effects of MMP3 knockdown in combination with cisplatin and compared the biological effects of MMP3 suppression and MMP3 inhibition. To identify the downstream MMP3 effectors that contribute to cisplatin resistance in HGSOC, we performed an RNA-seq analysis of MMP3-siRNA-treated cisplatin-resistant HGSOC cells in the presence or absence of cisplatin. We also used immunoprecipitation (IP) coupled with mass spectrometry (MS) to identify MMP3 binding partners that promote cisplatin resistance. Finally, we assessed the effect of multiple injections of a liposomal MMP3-siRNA formulation combined with cisplatin on tumor growth in an OC mouse model. Overall, our results indicate that increased MMP3 levels contribute to cisplatin resistance in OC, highlighting the need to further investigate the mechanisms of MMP3 regulation in tumor cells and their microenvironment.

## 2. Results

### 2.1. MMP3 Is Upregulated in Cisplatin-Resistant OC Cells Compared with Their Cisplatin-Sensitive Counterparts

We recently reported that MMP3 is post-transcriptionally regulated by miR-18a, a microRNA with a tumor-suppressive role in cisplatin-resistant OC cells [31]. Additionally, we observed higher MMP3 protein levels in cisplatin-resistant cells than in cisplatin-sensitive cells [31]. As shown in Figure 1, we confirmed that MMP3 levels were increased at the mRNA (Figure 1A) and protein levels (Figure 1B,C) in cisplatin-resistant cells compared to cisplatin-sensitive cells. OVCAR3/OVCAR3CIS are HGSOC cells (RRID:CVCL_0465). We generated OVCAR3CIS cells by exposing OVCAR3 cells to increasing doses of cisplatin [34]. Although A2780/A2780CP20 cells were isolated from an ovarian endometrioid adenocarcinoma tumor, they have been extensively used to understand the mechanisms of cisplatin resistance [35,36,37,38]. To quantitatively assess the intracellular MMP3 protein levels, we used a MMP3 Human ELISA (Enzyme-Linked Immunosorbent Assay) kit. As shown in Figure 1D, we observed higher levels of MMP3 in the cisplatin-resistant cells than in the cisplatin-sensitive cells. Intracellular MMP3 protein levels in cisplatin-resistant cells were roughly 3–4 times higher than those in cisplatin-sensitive cells (*** *p* < 0.0042).

#### 2.1.1. *MMP3 Activity Is Not Correlated with the Cisplatin Sensitivity of OC Cells*

We performed an MMP3 activity assay to measure the enzyme’s ability to degrade an MMP3-specific FRET (fluorescence resonance energy transfer) substrate. MMP3 intracellular activity was slightly lower in A278CP20 cells but unchanged in OVCAR3CIS cells compared to their cisplatin-sensitive counterparts (Figure 1E). Additionally, no change in MMP3 extracellular activity was observed in chemoresistant cell lines compared to their counterparts (Figure 1F). Next, we assessed MMP3 activity inhibition in OVCAR3CIS cells using two small-molecule inhibitors targeting the MMP3 catalytic domain [18,19]. We focused on OVCAR3CIS cells, as they are HGSOC, the most clinically relevant OC subtype. The IC50s of OVCAR3/OVCAR3CIS and A2780/A2780CP20 have been previously reported [39]. 

An MMP3 inhibitor screening assay revealed that Inhibitor I (UK 356618) effectively inhibited 50% or more of the MMP3 activity at concentrations ranging from 10 to 50 nM, while Inhibitor II (C21H23N7O2S2) did not reduce MMP3 activity at doses up to 50 μM (Figure 1G). Figure 1H shows that neither of the two inhibitors decreased OVCAR3CIS cell viability at concentrations ranging from 0.1–100 nM for Inhibitor I or 1–100 μM for Inhibitor II. These results suggest that MMP3 catalytic activity is not involved in the sensitivity of OC cells to cisplatin treatment.

#### 2.1.2. siRNA-Mediated Knockdown of MMP3 and Small-Molecule Inhibitors Have Different Effects on Viability, Proliferation, and Invasion of OC Cells

We next compared the biological consequences (cell viability, proliferation, and invasion) of targeting MMP3 with siRNA versus small-molecule inhibitors in OC cells. First, we verified the specificity of the siRNA against MMP3, ensuring that it did not inhibit other members of the MMP protein family, including MMPs 1, 2, 8, 9, 10, 13, and 16. OVCAR3CIS cells were transfected with MMP3-targeted siRNA, followed by Western blot analysis using specific antibodies against each MMP (Figure 2A). A dose-response experiment with cisplatin in OVCAR3CIS cells pre-treated with 50 nM MMP3-targeting siRNA showed a significant reduction in cell viability (*p* ≤ 0.0001) compared to that in NC-siRNA-treated cells (Figure 2B). Cisplatin concentrations were selected based on previous studies [31,39] and our dose-response curves. Similar results were observed in A2780CP20 cells (Figure 2C). However, MMP3 inhibitors did not enhance the sensitivity of the cells to cisplatin treatment (Figure 2D).

MMP3 inhibitors were also unable to reduce colony formation in OVCAR3CIS cells in the presence or absence of cisplatin (2µM), compared to untreated or DMSO-treated cells (Figure 2E and Appendix A). Conversely, MMP3-targeting siRNAs effectively reduced the number of OVCAR3CIS colonies compared to NC-siRNA by nearly 50% (*** *p* < 0.0005, Figure 2F and Appendix A). When co-treated with cisplatin, siMMP3 continued to reduce OVCAR3CIS colony formation by almost half compared to cisplatin alone or cisplatin plus NC-siRNA (*** *p* < 0.0005).

We also assessed the effect of MMP3 knockdown on OVCAR3CIS invasion ability. Invasion assays revealed that similar to the outcomes of cell viability and clonogenicity, MMP3 inhibitors failed to reduce the invasiveness of OVCAR3CIS cells in the presence or absence of 10µM cisplatin (Figure 2G and Appendix A). In contrast, MMP3-targeting siRNA significantly decreased the invasiveness of the cells in the presence of cisplatin compared to cisplatin plus NC-siRNA (37% reduction; ** *p* < 0.001). Notably, no differences were detected in OVCAR3CIS invasion between the MMP3-siRNA and NC-siRNA groups (Figure 2H and Appendix A).

We also assessed changes in MMP3 mRNA levels in OVCAR3CIS cells. As depicted in Figure 2I, treatment of cells with cisplatin resulted in significant upregulation of MMP3 expression compared to that in untreated (NT) cells (** *p* ≤ 0.005). Furthermore, we observed a marked decrease in MMP3 expression in cells treated with siMMP3 compared to that in NC-siRNA-treated cells (*** *p* ≤ 0.0004), confirming the efficacy of siRNA-mediated knockdown. Together, these results suggest that regions other than the catalytic domain are associated with the sensitivity of OC cells to cisplatin treatment.

### 2.2. Downstream Effectors of MMP3 in HGSOC Cells

The specific downstream MMP3 effectors that contribute to cisplatin resistance in OC cells have not been investigated. Thus, we carried out a transcriptome-wide analysis using RNA sequencing (RNA-seq) after siRNA-mediated MMP3 knockdown with and without cisplatin (50 nM) in OVCAR3CIS cells. Using an initial *p*-adjusted value cutoff of <0.01, we identified 113 differentially expressed genes (DEGs) between NC-siRNA- and cisplatin-treated OVCAR3CIS cells, 415 DEGs between NC-siRNA- and MMP3-siRNA-treated cells, and 440 DEGs exhibited differential expression between cells treated with NC-siRNA plus cisplatin and those treated with MMP3-siRNA plus cisplatin (Figure 3A). Notably, 144 DEGs were exclusive to MMP3-siRNA compared to NC-siRNA (Figure 3A). The full list of the top 50 DEGs for each condition can be found in Appendix A; 101 DEGs were exclusive to cisplatin treatment (Appendix A), and four DEGs were consistently differentially expressed across all comparisons (Figure 3A and Appendix A).

Functional enrichment analysis via Metascape using Gene Ontology and KEGG revealed that the enriched ontology clusters for the NC versus cisplatin conditions included calcium ion binding, pre-NOTCH transcription and translation, and integrin cell surface interactions (Figure 3B).

The enriched ontology clusters for the NC and siMMP3 conditions included the regulation of transmembrane transport, export from the cell, regulation of secretion, and extracellular matrix organization (Figure 3C). In contrast, the NC plus cisplatin versus siMMP3 plus cisplatin conditions included potassium and aquaporin channel-mediated transport deregulation, clathrin-mediated endocytosis, the cell cycle, metabolic processes, cellular response to stress, and extracellular matrix organization (Figure 3D).

To visualize the molecular interactions among the deregulated transcripts, the list of 577 transcripts from siMMP3-treated cells, with and without cisplatin (Figure 3A; 144+262+171 DEG’s), compared to NC was analyzed using Ingenuity Pathway Analysis (IPA), which identified the top 10 distinct networks. The top networks included genes involved in key molecular mechanisms related to cancer, including potassium channels and several signaling pathways. The notable genes and pathways identified include protein Wnt-11 (WNT11), the phosphoinositide-3-kinase (PI3K) complex, the extracellular signal-regulated kinase 1/2 (ERK1/2) cascade, mitogen-activated protein kinase kinase (MAP2K1/2), and potassium voltage-gated channel subfamily H member 2 (KCNH2). Additionally, copper-transporting ATPase 1 (ATP7A), which is known to facilitate the efflux of cisplatin from cells, was downregulated following MMP3 knockdown. Vascular endothelial growth factor C (VEGFC), potassium sodium-activated channel subfamily T member 2 (KCNT2), and mitogen-activated protein kinase 1 (MAPK1) were also identified, and their roles in molecular and vesicle trafficking, as well as apoptosis and proliferation mechanisms, were determined (Appendix A). Table 1 lists the top 20 DEGs (according to adjusted *p*-values) associated with siRNA-mediated MMP3 knockdown, the MMP3-dependent cisplatin response, and siRNA-mediated MMP3 knockdown plus cisplatin compared with NC-siRNA plus cisplatin.

### 2.3. MMP3 Is Associated with Different Proteins in Untreated and Cisplatin-Treated OVCAR3CIS Cells

MMP3 was successfully immunoprecipitated from serum-free OVCAR3CIS cisplatin-treated and untreated cells and their supernatants. Successful immunoprecipitation of MMP3 was confirmed by Western blotting (Appendix A).

Following this validation, we performed a proteomic analysis of the immunoprecipitated samples. In untreated OVCAR3CIS cells, 105 intracellular proteins co-immunoprecipitated (co-IPed) with MMP3, 27 of which were consistent across at least two replicates. Conversely, in cisplatin-treated OVCAR3CIS cells, 97 intracellular proteins co-IPed with MMP3, 38 of which were consistent in at least two replicates. In both cisplatin-treated and untreated OVCAR3CIS cells, 22 proteins co-IPed with intracellular MMP3. Additionally, five proteins were unique to the untreated cells, and 16 proteins were unique to the cisplatin-treated OVCAR3CIS cells (Figure 3E). The proteins co-IPed with intracellular MMP3 are listed in Table 2.

With respect to extracellular MMP3, in untreated OVCAR3CIS cells, 29 co-IPed proteins were identified, whereas in cisplatin-treated cells, 21 co-IPed proteins were observed. Under both conditions, eight proteins co-IPed with extracellular MMP3. Additionally, 21 proteins were unique to untreated cells, and 13 proteins were unique to cisplatin-treated OVCAR3CIS cells (Figure 3F).

When the proteins that co-IPed with intracellular and extracellular MMP3 were compared, 16 proteins were exclusive to intracellular MMP3, whereas 12 were unique to extracellular MMP3 (Figure 3G). The proteins co-IPed with extracellular MMP3 are listed in Table 3. Interestingly, only one protein, coiled-coil domain-containing protein 22 (CCDC22), was co-IPed with intra- and extracellular MMP3 (Figure 3G). These data indicate that MMP3 interacts with different proteins in untreated and cisplatin-treated OVCAR3CIS cells. The identification of proteins co-IPed with MMP3 suggests their potential role in enhancing the survival of cisplatin-resistant OC cells.

### 2.4. Effect of Multiple Injections of Liposomal-siRNA-MMP3 in an OC Mouse Model

Next, we aimed to determine the therapeutic effects of targeting MMP3 with liposomal-siRNA in an OC mouse model. We encapsulated the siRNAs into DOPC-based nanoliposomes. The characterization of these liposomal formulations has been previously described [40]. Notably, there were no significant differences in mouse weights between the groups (Figure 4A). Additionally, the number of nodules, with and without cisplatin administration, did not differ significantly between the NC-siRNA and MMP3-siRNA groups or between the NC-siRNA plus cisplatin and MMP3-siRNA plus cisplatin groups (Figure 4B). Surprisingly, multiple injections of liposomal MMP3-siRNA increased, although not statistically significant, tumor weight in the siMMP3 group compared to the negative control (NC) group. However, a significant reduction in tumor growth was observed in the MMP3-siRNA plus cisplatin group compared to the NC-siRNA plus cisplatin group (* *p* < 0.05, Figure 4C).

We also assessed whether the liposomal MMP3-siRNA injection reduced MMP3 protein expression levels. Compared to those treated with NC liposomes, the MMP3 protein levels in the mice treated with siMMP3 liposomes were markedly lower (Figure 4D). Interestingly, compared with those in the siMMP3 group without cisplatin, MMP3 expression levels in mice treated with siRNA liposomes plus cisplatin were elevated. However, we observed a notable decrease in MMP3 expression in mice treated with siMMP3 liposomes plus cisplatin compared to those treated with NC-siRNA liposomes and cisplatin (Figure 4D). In summary, our findings highlight that combination therapy involving liposomal MMP3-siRNA and cisplatin effectively mitigates tumor growth in a cisplatin-resistant mouse model of HGSOC but raises concerns about the feasibility of targeting MMP3 alone in patients with ovarian cancer.

### 2.5. In Vivo Targeting of MMP3 with Liposome-Encapsulated siRNAs Reduces Cell Proliferation and Angiogenesis in an OC Mouse Model

We investigated the biological effects of MMP3-siRNA therapy on tumor cell proliferation and blood vessel formation. Compared with those treated with NC-siRNA or NC plus cisplatin, mice treated with MMP3-siRNA alone (**** *p* ≤ 0.0001) or in combination with cisplatin (* *p* ≤ 0.05) showed significant decreases in cell proliferation, as evidenced by Ki-67 immunostaining (Figure 4E,F). Furthermore, we evaluated the effects of liposomal MMP3-siRNA on angiogenesis by assessing CD31 expression. Notably, compared with those treated with liposomal NC-siRNA or NC-siRNA plus cisplatin, mice treated with liposomal MMP3-siRNA alone (*** *p* ≤ 0.0001) or in combination with cisplatin (* *p* ≤ 0.05) showed a significant decrease in CD31 staining (Figure 4E–G). Furthermore, we assessed the effectiveness of MMP3-siRNA in the tumor tissues of mice. Compared with that in the control groups, MMP3 expression in the siMMP3-treated mice, either alone (* *p* ≤ 0.01) or in combination with cisplatin (** *p* ≤ 0.01), was significantly lower. Interestingly, MMP3 expression levels were higher in mice treated with cisplatin than in those not treated with cisplatin (Figure 4E–H).

### 2.6. Expression of MMP3 in Human Ovarian Cancer Patients

To assess the clinical relevance of MMP3 in OC, we interrogated the KM plotter patient database. As illustrated by the Kaplan-Meier curves depicted in Figure 5, elevated levels of MMP3 were found to significantly diminish both progression-free survival (PFS) (Figure 5A, *p* ≤ 0.002) and overall survival (OS) (Figure 5B, *p* ≤ 0.025) in OC patients. These findings strongly suggest an association between MMP3 expression and survival outcomes in patients with OC.

## 3. Discussion

The main clinical challenge for women with HGSOC is acquired resistance to cisplatin, which can lead to recurrence, treatment failure, and, ultimately, death. This highlights the urgent need for more effective treatments for advanced and drug-resistant tumors. Our study provides novel evidence for the role of MMP3 in cisplatin-resistant OC and its potential as a therapeutic target. Notably, our findings suggest that MMP3 catalytic activity is not required for its role in chemoresistance, indicating a non-proteolytic mechanism. Furthermore, we provide critical insights into MMP3-associated protein networks and downstream effectors through proteomic and transcriptomic analyses.

Cisplatin exposure leads to MMP3 promoter demethylation, enhancing chemoresistance, tumor growth, and metastasis [10,41]. Additionally, high MMP3 levels correlate with poor chemotherapy response in various cancers, suggesting that targeting MMP3 could improve therapeutic outcomes [10,42]. The clinical relevance of MMP3 is further supported by our findings using the KM plotter patient database, which revealed that elevated MMP3 levels are associated with poorer overall survival and progression-free survival in patients with OC. In the present study, we observed increased mRNA and protein levels of MMP3 in cisplatin-resistant OC cells compared to those in their cisplatin-sensitive counterparts. Knockdown of MMP3 significantly decreased the proliferation, viability, and invasion of cisplatin-resistant OC cells, with even greater effects when MMP3 knockdown was combined with cisplatin treatment. This highlights the contribution of MMP3 to cisplatin resistance in OC. However, the use of commercially available MMP3 chemical inhibitors did not reproduce the biological effects observed with siRNA-mediated knockdown.

RNA-seq revealed differential gene expression patterns associated with siRNA-mediated knockdown of MMP3. Functional enrichment analysis via Metascape using Gene Ontology and KEGG revealed that enriched ontology clusters included genes involved in potassium and aquaporin channel-mediated transport deregulation, metal ion export, clathrin-mediated endocytosis, the cell cycle, metabolic processes, cellular response to stress, and extracellular matrix organization. Together, these genes and processes highlight the complex interplay between cell cycle regulation, apoptosis, cell adhesion, cell transport, and metabolism in the development of chemoresistance. These processes may contribute to cisplatin resistance in HGSOC.

Critical genes associated with MMP3 function include the secretory carrier membrane protein 5 (SCAMP5), which promotes calcium-triggered cytokine secretion by interacting with the SNARE (“SNAP Receptors”) machinery. SNARE affects the internalization, trafficking, and fate of receptors, such as EGFR, which are known to be linked to cisplatin resistance pathways and alter the response of cancer cells to cisplatin-induced DNA damage [43,44].

Secretogranin III (SCG3), notably upregulated in the NC plus cisplatin group compared to the siMMP3 plus cisplatin group, is a member of the granin protein family that regulates the biogenesis of secretory granules, serving as storage compartments for secretory products and functioning as angiogenic factors under pathological conditions [45]; secretogranin-1 (CHGB), in the same group, is another protein associated with secretory granules. CHGB plays a critical role in regulating secretion by forming highly selective anion channels in the granule membranes during exocytosis [46]. Another significantly upregulated gene included synaptic vesicle 2-related protein (SVOP), a member of the solute carrier family 22 that transports toxins and drugs [47].

The genes identified in the NC-siRNA plus cisplatin group versus the MMP3-siRNA plus cisplatin group included cell division cycle 25A (CDC25A), F11 receptor (F11R), actin-related protein 2 (ACTR2), and ADAM metallopeptidase with thrombospondin type 1 motif like 4 (ADAMTSL4). CDC25A is a dual-specificity protein phosphatase that plays a crucial role in cell cycle regulation by removing inhibitory phosphorylation from cyclin-dependent kinases, thereby promoting cell cycle progression. Overexpression of CDC25A is commonly observed in various cancers and is frequently linked to poor prognosis and resistance mechanisms through its regulation of the cell cycle and apoptosis [48]. 

The differential effects observed with MMP3-targeted siRNA versus small-molecule inhibitors emphasize the importance of targeting MMP3 in regions other than the catalytic domain. When we compared the intracellular and extracellular MMP3 activity levels between resistant and sensitive OC cell lines, our results revealed slightly lower intracellular MMP3 activity levels in the resistant OC cell lines than in their sensitive counterparts, with no significant changes in extracellular activity levels. These findings suggest that neither extracellular nor intracellular MMP3 activity contributes to the chemoresistant phenotype of these cell lines. Evidence indicates that MMPs can execute roles outside of their proteolytic activity [14,49,50]. By engaging with signaling receptors, the hemopexin domain of secreted MMPs may trigger signaling cascades involved in disease pathology. The MMP3 hemopexin domain has a large surface area that can act as a scaffold, promoting different protein-protein interactions [49,50,51]. Previous studies by Kessenbrock et al. have shown that MMP3 regulates the Wnt signaling pathway by binding to and inactivating Wnt5b via the hemopexin domain, controlling mammary stem cell function [52]. Notably, several cancer-related Wnt proteins function as extrusion pumps, expelling chemotherapeutic drugs from cell [53]. For example, the drug extrusion pump MDR-1 (P-GP, ABCB1) and the CD44 family of cell adhesion molecules are among the Wnt targets involved in drug resistance [53]. Moreover, Correia et al. reported that the MMP3 hemopexin domain interacts with the chaperone HSP90β and that this interaction is required for mouse mammary epithelial cell invasion [54]. HSP90β has also been implicated in multidrug resistance in OC via AKT/GSK3β/β-catenin signaling [55].

To identify MMP3-interacting proteins in cisplatin-resistant OC, we performed IP coupled with MS. To the best of our knowledge, this is the first study to identify the intracellular interactions of MMP3 associated with cisplatin resistance in OC. Particularly, ALB, CCDC22, SRXN1, CLTC, and ATXN2 are involved in vesicle trafficking, DNA repair, stress response, transcriptional regulation, extracellular matrix remodeling, signal transduction, immune interactions, and cellular integrity. These proteins likely contribute to the complex network of pathways that increase the survival of ovarian cancer cells resistant to cisplatin. Understanding their precise roles and interactions could provide new insights into overcoming chemoresistance and improving therapeutic outcomes in patients with OC.

For example, plasma albumin (ALB) can strongly bind to cisplatin, leading to the inactivation of a large amount of cisplatin and significantly reducing the amount that directly enters cells to form DNA adducts and exerts its cytotoxic effects [56]. The reactions of cisplatin with ALB are thought to play an important role in the metabolism of this anticancer drug [56]. In our study, ALB was found to coimmunoprecipitate (co-IP) with MMP3 intracellularly only upon cisplatin treatment, whereas extracellular co-IP was observed in both untreated and cisplatin-treated cells. These findings indicate that cisplatin treatment may facilitate the internalization of the albumin-MMP3 complex or alter cellular conditions to promote their interaction within cells. The results of the intracellular co-IP of ALB with MMP3 upon cisplatin treatment suggest a potential role for MMP3 in modulating cisplatin sensitivity. By binding to ALB, MMP3 may influence the sequestration and internalization of cisplatin–albumin complexes, thereby affecting the intracellular availability of cisplatin. This interaction could reduce the formation of DNA adducts and contribute to cisplatin resistance.

We identified chromosome 18 open reading frame 63 (C18orf63) as a protein that co-IP with MMP3 intracellularly after cisplatin treatment. C18orf63, the function remains largely uncharacterized; its interaction with MMP3 may indicate a role in chromosomal or transcriptional regulation. Another protein observed to co-IP intracellularly with MMP3 upon cisplatin treatment was H2BC21, a histone that plays a role in chromatin structure and gene regulation, suggesting that MMP3 may influence transcriptional regulation through its interaction with histones. H2BC21 has also been linked to poor prognosis in gliomas and is associated with tumor immunity [57]. Histones H4 and H2AC20 were identified as additional intracellular MMP3 binding partners in both cisplatin-treated and untreated OVCAR3CIS cells.

We observed that ribosomal proteins (RPL27A, RPL36AL, RPS15, RPS23, RPS29, RPS4Y2, and RPS6) co-immunoprecipitated with MMP3 intracellularly after cisplatin treatment. These ribosomal proteins are essential for protein synthesis, and their interaction with MMP3 suggests a potential influence on translation regulation, which could be an adaptation to the high protein synthesis demands of rapidly dividing cancer cells. Additionally, heat shock protein family A member 9 (HSPA9) was found to co-IP with MMP3. HSPA9 is involved in protein folding and mitochondrial function, indicating a potential role for MMP3 in maintaining cellular homeostasis under stress conditions, such as cisplatin treatment. Notably, HSPA9, also known as GRP75, controls cisplatin resistance in patients with ovarian cancer by facilitating the integrity of the mitochondria-associated ER membrane (MAM) [58]. The interaction between MMP3 and HSPA9 can impact cancer therapy by influencing ER-mitochondrial calcium flux, cell fate under stress, and mitochondrial function. Sulfiredoxin-1 (SRXN1) is also co-immunoprecipitated with MMP3 intracellularly after cisplatin treatment. SRXN1 plays a role in reducing oxidative stress by repairing peroxiredoxins, and its interaction with MMP3 may play a role in managing the oxidative environment within cancer cells.

Recent studies have shown that histones can be actively secreted via extracellular vesicles (EVs)/exosomes, particularly under cellular stress conditions [59]. These extracellular histones, often membrane-bound rather than nucleosomal, have been detected in biofluids and tumor microenvironments, where they may contribute to cancer progression and drug resistance mechanisms. Similarly, ribosomal proteins, traditionally linked to intracellular functions, have been implicated in extracellular signaling and RNA transport via exosomes [60]. Notably, in our study, we identified histones and ribosomal proteins extracellularly, suggesting that these proteins may be secreted alongside MMP3. Given that MMP3 is associated with exosome-mediated secretion, the co-immunoprecipitated proteins identified in our study may have been secreted via EVs or other unconventional pathways.

CCDC22, the only protein identified as a common MMP3 intra- and extracellular binding partner in OVCAR3CIS cisplatin-treated cells, is involved in the regulation of NF-kappa-B signaling [61]. Interestingly, NF-κB binds to the MMP3 gene promoter, inducing the expression of MMP364. CCDC22 can act as a bridge between MMP3 and the NF-κB complex, which could enhance NF-κB binding to the MMP3 promoter, leading to increased MMP3 expression. Alternatively, CCDC22 may act as a negative regulator of NF-κB binding to the MMP-3 promoter. These co-IP results indicate that CCDC22 sequesters MMP3, preventing its interaction with NF-κB, thereby reducing MMP3 expression. CCDC22 may also regulate the stability or localization of MMP3, which could indirectly influence its interaction with NF-κB. CCDC22 is also a component of the CCC complex, which is involved in the regulation of endosomal recycling of surface proteins, including integrins, signaling receptors, and channels, and also plays a role in copper ion homeostasis [62,63]. The co-IP of CCDC22 with MMP3 in cisplatin-resistant OVCAR3CIS cells highlights a potentially significant interaction that may contribute to cisplatin resistance in ovarian cancer cells.

The identification of these MMP3 binding partners in cisplatin-resistant OVCAR3CIS cells revealed that different proteins are involved in vesicle trafficking, DNA repair, stress response, transcriptional regulation, ECM remodeling, signal transduction, immune interactions, and cellular integrity. These proteins likely contribute to the complex network of pathways that increase the survival of ovarian cancer cells resistant to cisplatin. Understanding their precise roles and interactions could provide new insights into overcoming chemoresistance and improving therapeutic outcomes in patients with ovarian cancer.

Liposomal MMP3-siRNA was used to enhance the stability, specificity, and delivery efficiency of siRNA in ovarian cancer cells. Nanoparticle-based carriers, particularly liposomes, protect siRNAs from degradation, facilitate targeted delivery, and reduce toxicity, thereby addressing key challenges such as rapid degradation and off-target effects. In the context of cisplatin-resistant ovarian cancer, targeting MMP3 with siRNA offers a promising therapeutic approach. Liposomal formulations ensure effective delivery to the tumor microenvironment, minimizing exposure to healthy cells while maximizing gene silencing in resistant cancer cells. This strategy aims to counteract MMP3-driven chemoresistance, potentially improving the treatment outcomes of HGSOC.

While liposomal MMP3-siRNA alone did not significantly reduce tumor size, it lowered cell proliferation and angiogenesis. This suggests that MMP3 knockdown affects tumor biology but is not sufficient on its own to shrink tumors within the study timeframe. However, when combined with cisplatin, the synergistic effect led to a significant reduction in tumor growth, indicating that MMP3 inhibition enhances cisplatin sensitivity rather than acting as a standalone treatment. Notably, cisplatin treatment increased MMP3 expression in the tumors of these mice, highlighting the critical role of MMP3 in regulating cisplatin resistance in OC. MMP3 is regulated by complex interactions within the tumor microenvironment (TME) [64,65]. Previous studies have shown that MMP3, which is mostly secreted by macrophages, neutrophils, and fibroblasts, interacts with various cells in the TME, such as cancer-associated fibroblasts (CAFs), tumor-associated neutrophils (TANs), tumor-associated macrophages (TAMs), and other immune cells, to promote tumor progression and resistance mechanisms [66,67]. For example, MMP3 has been found to enhance the pro-tumorigenic activities of CAFs, leading to increased tumor cell invasion and metastasis [68,69]. Additionally, MMP3 can modulate the immune response, creating an immunosuppressive environment that allows tumor cells to evade immune detection [64]. CAFs can contribute to MMP3 upregulation in cancer cells by secreting factors that influence MMP3 expression and remodeling the extracellular matrix, further influencing MMP3 activity [64]. These interactions may explain why the knockdown of MMP3 with siRNA alone did not significantly reduce tumor growth in our study. Further research is needed to clarify the role of MMP3 in the TME of OC tumors.

## 4. Methods

### 4.1. Cell Lines and Cell Culture Maintenance

The human epithelial ovarian cancer cell line A2780 (RRID:CVCL_0134) and the HGSOC cell line OVCAR3 (RRID:CVCL_0465) were purchased from the American Type Culture Collection (ATCC; Manassas, VA, USA). A2780CP20 (a cisplatin-resistant subline of A2780) was kindly provided by Dr. Anil K. Sood. OVCAR3CIS cells were generated by exposing parental cell lines to increasing doses of cisplatin, as previously described [34]. OVCAR3 and OVCAR3CIS cells were maintained in RPMI-1640 (Thermo Scientific, Mount Prospect, IL, USA) medium supplemented with 0.01 mg/mL insulin (Sigma-Aldrich, St. Louis, MO, USA), while A2780 and A2780CP20 cells were maintained in RPMI-1640 medium without insulin. In all cases, the medium was supplemented with 10% fetal bovine serum (FBS; HyClone, GE Healthcare Life Sciences, Logan, UT, USA) and 0.1% antibiotic/antimycotic solution (HyClone). All cells were maintained in a humidified incubator at 37 °C in 5% CO_2_ and 95% air. Cell lines were screened for mycoplasma using the LookOut^®^ Mycoplasma PCR detection kit as described by the manufacturer (Sigma-Aldrich) and authenticated by Promega (Madison, WI, USA) and ATCC using Short Tandem Repeat analysis. In vitro experiments were performed using cells at 70–85% confluence.

### 4.2. Media Concentration

Media from the respective cell lines were concentrated using 3kDa MWCO Amicon^®^ Ultra15 Centrifugal Filter Units (EMD Millipore, Burlington, MA, USA) with a cutoff of 3000 Da and centrifuged at 7500 rpm for 40 min. Total protein content was subsequently quantified.

### 4.3. Quantitative Real-Time PCR (qRT-PCR)

RNA was isolated from the cell pellets using the GenElute Mammalian Total RNA Isolation Kit (Sigma-Aldrich), according to the manufacturer’s instructions. Complementary DNA (cDNA) was synthesized using an iScript cDNA synthesis kit (Bio-Rad Laboratories, Inc., Hercules, CA, USA) with 1.0 μg of total RNA as the starting material. qPCR was carried out in a StepOne Plus thermal cycler system (Applied Biosystems, Waltham, MA, USA) with the following program: 10 min at 95 °C, 40 cycles of 15 s at 95 °C, and 1 min at 60 °C. The gene-specific primers used were as follows: MMP3 forward, 5′-TGAAATTGGCCACTCCCTGG-3′; and MMP3 reverse, 5′-GGAACCGAGTCAGGTCTGTG-3′. The gene expression level was defined as the threshold cycle number (Ct). The mean fold changes in the expression of the target genes were calculated using the comparative ΔΔCt method relative to the sensitive cell lines and normalized to β-actin expression using the primers β-actin forward (5′-AGAGCTACGAGCTGCCTGAC-3′) and β-actin reverse (5′-AGCACTGTGTTGGCGTACAG-3′).

### 4.4. Western Blot Analysis

Cell pellets were lysed with ice-cold lysis buffer (1% Triton X, 150 mmol/L NaCl, 25 mmol/L Tris HCl, 0.4 mmol/L NaVO_4_, 0.4 mmol/L NaF, and protease inhibitor cocktail (Sigma)) and incubated on ice for 30 min. Whole-cell lysates were centrifuged, and the supernatants were collected for further analysis. Protein concentrations were measured using the Pierce BCA protein assay kit protein assay (Thermo Scientific Waltham, MA, USA). Proteins (30–50 μg) were separated via SDS-PAGE (12% acrylamide), blotted onto nitrocellulose membranes, and probed overnight at 4 °C with primary antibodies targeting MMP3 (ab52915; 1:1000; Abcam, Cambridge, UK), MMP1 (PB9725; 1:1000; Boster, Pleasanton, CA, USA), MMP2 (A00286; 1:1000; Boster), MMP8 (PB9726; 1:1000; Boster), MMP9 (PB9668; 1:1000; Boster), MMP10 (PB9670; 1:1000; Boster), MMP13 (A00420–2; 1:1000; Boster) and MMP16 (PA1123; 1:1000; Boster;). β-actin (A5441; 1:5000; Millipore-Sigma, St. Louis, MO, USA) and Ponceau S staining were used as the loading controls. The membranes were rinsed and incubated with the corresponding horseradish peroxidase-conjugated secondary antibody (Cell Signaling Technology, Danvers, MA, USA), followed by enhanced chemiluminescence (GE Healthcare, Chicago, IL, USA) and autoradiography using a ChemiDoc Imaging System (Bio-Rad, Hercules, CA, USA).

### 4.5. In Vitro MMP3-siRNA Transfection and Treatment with MMP3 Inhibitors

OVCAR3CIS cells (3.0 × 10^4^ cells/mL) were seeded the day before transfection. Cells were transfected with a previously validated siRNA targeting MMP3 [31] (siMMP3:5′-CACAATATGGGCACTTTAA-3′) or scrambled negative control siRNA (NC-siRNA: 5′-GACCGCGAATAGACGAACG-3′). A final concentration of 50 nM siRNA was mixed with HiPerFect transfection reagent (Qiagen, Germantown, MD, USA) at a 1:2 ratio (siRNA:HiPerFect) in serum- and antibiotic-free Opti-MEM medium (Thermo Fisher Scientific, Waltham, MA, USA). The transfected cells were incubated overnight, and cell pellets were collected the following day for subsequent experiments.

We also used two commercially available MMP3 pharmacological inhibitors: Inhibitor I, UK-356618 (Millipore-Sigma) [70], and Inhibitor II, C27H46N10O9S (Santa Cruz Biotechnology, Dallas, TX, USA) [71]. Fresh batches of inhibitors were prepared every two months to ensure proper activity. Similarly, cisplatin (Sigma-Aldrich) stock solutions were prepared every two months, filtered, and stored until required.

### 4.6. MMP3 Activity Assay

An MMP3 activity assay kit (ab118972; Abcam) was used to measure MMP3 activity in cisplatin-sensitive and cisplatin-resistant OVCAR3 and A2780 cells and cell culture media. A2780, A2780CP20, OVCAR3, and OVCAR3CIS cells were cultured in clear 96-well plates at a density of 3.0 × 10^4^ cells/mL. To directly measure MMP3 activity, the cells and supernatants were transferred to a black 96-well plate. After reacting with the MMP3 substrate (prepared according to the protocol), the plate was read at Ex/Em = 325/393 nm twice for 2 h.

### 4.7. Cell Viability Assays

For cell viability assays, 3.0 × 10^4^ cells/mL were seeded in 96-well plates and transfected with siRNAs or treated with MMP3 inhibitors, either alone or in combination with cisplatin at concentrations ranging from 0.1 to 100 µM. Seventy-two hours post-transfection or treatment, Alamar blue dye (Thermo Fisher Scientific) was added, and after 3 h of incubation at 37 °C, absorbance values at 570 nm were measured using a spectrophotometer (Bio-Rad). Cell viability was calculated as a percentage by normalizing the absorbance values to those of the untreated control cells.

### 4.8. Clonogenic Assays

To assess cell growth and proliferation, clonogenic assays were performed using crystal violet dye [34]. Briefly, OVCAR3CIS cells were seeded into 12-well plates, and 24 h later, they were transfected with 50 nM of MMP3-targeting or negative control (NC) siRNAs. The cells were treated with 2 µM cisplatin 24 h post-transfection. The next day, the transfected cells were seeded in 10-cm Petri dishes and incubated for seven days. After incubation, the colonies were fixed and stained with 0.5% crystal violet in methanol. Colonies of at least 50 cells were quantified using light microscopy at 10× magnification in five random fields. To compare the effects of siMMP3 and MMP3 small-molecule inhibitors on cell growth and proliferation, we conducted a clonogenic assay on OVCAR3CIS cells treated with Inhibitor I (10 or 50 nM), Inhibitor II (10 or 50 μM), or DMSO in the presence or absence of cisplatin.

### 4.9. Migration Assays

For cell invasion, OVCAR3CIS cells (2 × 10^4^ cells/mL) were seeded in 10-cm tissue culture dishes. After 24 h, the cells were treated with MMP3 inhibitors or siRNAs, with or without 10 µM cisplatin. The next day, 70,000 cells were seeded into Matrigel-coated Transwell plates. Forty-eight hours later, the cells were fixed and stained using the Fisher HealthCare™ PROTOCOL™ Hema 3™ Manual Staining System (Thermo Fisher Scientific). Invading cells were counted at 20× magnification using an Olympus (Olympus, Center Valley, PA, USA) 1 × 71 microscope equipped with a digital camera (Olympus DP26). The percentage of invasion was calculated by normalizing to the values of untransfected cells.

### 4.10. RNA Sequencing of siRNA-Mediated MMP3-Knockdown Cells with and Without Cisplatin

To prepare the RNA sequencing library, total RNA was isolated from OVCAR3CIS cells treated with cisplatin alone, NC-siRNA, or MMP3-siRNA alone, or a combination of both treatments. RNA was also isolated from controls, including untreated cells and cells transfected with NC-siRNA (four biological replicates of each condition), using the Mirvana RNA Isolation Kit (Fisher, Grand Island, NY, USA). The integrity of the RNA samples was evaluated using an Agilent Bioanalyzer 2100, and 1.0 µg of high-quality RNA was used in the sequencing protocol. RNA was enriched, and the library was prepared via GENEWIZ^®^ Strand-specific RNA sequencing with rRNA depletion (GENEWIZ, Inc., South Plainfield, NJ, USA). The library was quantified using KAPA SYBR^®^ FAST qPCR and sequenced using the Illumina HiSeq platform (2 × 150 bp) (San Diego, CA, USA) with a sequencing depth of approximately 100 million reads per sample. Sequence reads were trimmed to remove possible adapter sequences and nucleotides of poor quality via Trimmomatic v.0.36. The trimmed reads were mapped to the *Homo sapiens* GRCh38 reference genome available on ENSEMBL using STAR aligner v.2.5.2b. Unique gene counts were calculated via featureCounts from the Subread package (version 1.5.2; Parkville, VIC, Australia), and initial gene expression analysis was performed using the DESeq2 (version 1.28.1) package in the R version 4.0.1 package.

### 4.11. Kyoto Encyclopedia of Genes and Genomes Pathway Enrichment, Gene Ontology, and Network Analyses

RNA transcripts exhibiting significant differences in abundance, with a log2-fold change cutoff beyond ±1.2 and a *p*-value ≤ 0.01, were selected for further investigation. The involvement of these transcripts in diverse biological pathways was explored using the Kyoto Encyclopedia of Genes and Genomes (KEGG) pathway enrichment analysis. Additionally, the enrichment of gene ontology terms encompassing biological processes, molecular functions, and cellular components was conducted via Metascape (https://Metascape.org/gp/index.html#/main/step1; accessed on 13 January 2023).

For network analysis, we used Ingenuity Pathway Analysis 24.0.2 (IPA; Ingenuity Systems, Qiagen, Redwood City, CA, USA) software to determine functional networks and pathways associated with differentially abundant RNA transcripts using a *p*-value cutoff of <0.01.

### 4.12. Immunoprecipitation of the MMP3 Protein

MMP3 was immunoprecipitated (IP) using a Pierce MS-compatible Magnetic IP Kit (Thermo Fisher Scientific) according to the manufacturer’s instructions. Briefly, the labeled protein mixture was pre-cleared against washed protein A/G magnetic beads (20 μL/200 μg of protein) at 4 °C for 1 h. After bead separation, the supernatant was collected and incubated with an anti-MMP3 antibody (Abcam) at 10 μg/200 μg of protein at 4 °C overnight. The samples were then added to pre-washed protein A/G magnetic beads (25 μL) for 1 h. The beads were collected and washed three times with IP-MS wash buffer. After washing, the beads were mixed with 100 μL of the elution buffer for 10 min. The supernatant was collected and dried for MS analysis.

### 4.13. MS Analyses and Protein Identification

Peptide separation was performed using an HPLC system (Easy nLC 1200, Thermo Fisher Scientific). Peptides were loaded onto a Pico Chip H354 REPROSIL-Pur C18-AQ 3 µM 120 A (75 µm × 105 mm) chromatographic column (Agilent, Santa Clara, CA, USA). The separated peptides were analyzed using a Q Exactive Plus mass spectrometer (Thermo Fisher Scientific) operated in positive polarity mode and data-dependent mode. MS1 (full scan) was measured over the range of 375–1400 *m*/*z* at a resolution of 70,000. MS2 (MS/MS) analysis was used to select the ten most intense ions for HCD fragmentation over the range of 200–2000 *m*/*z* at a resolution of 35,000. A dynamic exclusion parameter was set for 30.0 s with a repeat count of three times.

The MS/MS raw data files were searched against a forward and reverse human protein database from UniProt (version 2018) (Universal Protein Source) (www.uniprot.org). Protein identification was performed using Proteome Discoverer version 2.1.1 (Thermo Fisher Scientific) with the SEQUEST HT algorithm. The false discovery rate was set at 0.01 (strict) and 0.05 (relaxed) for the two groups. The raw protein files obtained from the Proteome Discoverer software were exported in. xls format using Microsoft Excel spreadsheet Program 2016.

### 4.14. OVCAR3CIS Tumor Implantation and MMP3-Targeted Treatment with Cisplatin

Female athymic nude mice (NCr-nu, 6 weeks old) were purchased from Taconic (Hudson, NY, USA). To assess the therapeutic efficacy of liposomal MMP3-siRNA alone or in combination with cisplatin in vivo, mice were intraperitoneally (i.p.) injected with OVCAR3CIS (1.2 × 10^6^ cells/0.2 mL HBSS). Seven days after cell implantation, the mice were randomly divided into the following treatment groups (n = 10 per group): (a) NC-siRNA, (b) MMP3-siRNA, (c) NC-siRNA plus cisplatin, and (d) MMP3-siRNA plus cisplatin. Liposomal siRNAs (5 µg siRNA/injection) and cisplatin (3 mg/kg) were injected (i.p.) twice a week for four weeks. At the end of the treatment, the mice were euthanized, the tumors were collected, and the number of tumor nodules and tumor weights were recorded. The sample size for the animal experiments was determined based on NIH guidelines to use the minimal number of mice required to achieve statistical significance, consistent with IACUC-approved protocols and published literature, where similar studies typically included 10 mice per group. The animal handling and research protocols used were approved by the Institutional Animal Care and Use Committee (IACUC) of the University of Puerto Rico, Medical Sciences Campus.

### 4.15. Immunohistochemistry

Immunohistochemistry was performed with specific antibodies against MMP3, CD31, and Ki-67, all purchased from Abcam, in paraffin-embedded tumor samples from the xenograft model experiments. Briefly, tissue slides were deparaffinized and rehydrated and then immersed in distilled water containing 3% hydrogen peroxide to inhibit endogenous peroxidase activity. Antigen retrieval was performed using microwave treatment for 15 min in an antigen unmasking solution (Vector Laboratories, Inc., Burlingame, CA, USA). The sections were incubated overnight at 4 °C with the specified antibodies, each diluted 1:100 in Dako antibody diluent (Dako North America Inc., Carpinteria, CA, USA). Afterward, the EnVision peroxidase-labeled polymer HRP (goat anti-rabbit, ready-to-use; Dako North America Inc.) was applied, and signals were developed using diaminobenzidine (DAB) chromogen (Dako North America Inc.). For quantification, three tissue areas were analyzed in four randomly selected mice per condition. Positively stained cells were quantified using ImageJ software v1.52, and immunoreactivity was estimated by counting the positively stained cells in the selected tissue areas.

### 4.16. Survival Analysis

Kaplan-Meier survival analysis was performed using publicly available patient datasets via the Kaplan-Meier (KM) plotter (www.kmplot.com). By selecting the MMP3 gene symbol, ovarian cancer patients were divided into high- and low-expression groups according to the median value of their RNA expression. A set of different filters was applied in our search, including those for patients with ovarian cancer, patients with ovarian cancer treated with platinum, patients with serous ovarian cancer, patients with stage 3 + 4 disease, patients treated with platinum, and patients with mutated TP53. Kaplan-Meier survival plots for overall survival (OS) and progression-free survival (PFS) were generated with hazard ratios (HRs), confidence intervals (CIs), and *p*-values (log ranks). In these studies, *p*-values < 0.05 were considered statistically significant.

### 4.17. Statistical Analysis

Statistical analyses and graph construction were performed using GraphPad Prism software (version 10.2.3; GraphPad Software Inc., La Jolla, CA, USA). Parametric methods (*t*-test or ANOVA) were used to calculate *p*-values as determined by normality tests. Statistical significance was set at *p* < 0.05.

## 5. Conclusions

Acquired resistance to platinum-based chemotherapy remains a primary clinical challenge in HGSOC, highlighting the urgent need for novel targeted therapies. Our study identified MMP3 as a key player in cisplatin resistance, demonstrating that MMP3 knockdown reduces cell viability, proliferation, and invasion in resistant ovarian cancer cells, with enhanced effects when combined with cisplatin. While small-molecule inhibitors targeting the MMP3 catalytic domain did not replicate these effects, our findings suggest that MMP3’s role in chemoresistance may be independent of its enzymatic activity and is likely mediated through protein-protein interactions. Through RNA-seq and proteomic analyses, we identified novel MMP3-interacting proteins involved in vesicle trafficking, DNA repair, metabolic adaptation, and stress response, further linking MMP3 to cisplatin resistance mechanisms. In vivo, MMP3-siRNA therapy alone did not significantly reduce tumor size; however, when combined with cisplatin, it resulted in a marked decrease in tumor growth, proliferation, and angiogenesis, reinforcing MMP3’s role in mediating chemoresistance. Additionally, our findings suggest that MMP3 expression is influenced by the tumor microenvironment (TME), which potentially modulates immune responses and stromal interactions. These results support MMP3 as a promising therapeutic target and emphasize the need for alternative inhibition strategies beyond its catalytic domain, which may improve treatment outcomes in chemoresistant ovarian cancer.

## Figures and Tables

**Figure 1 ijms-26-04012-f001:**
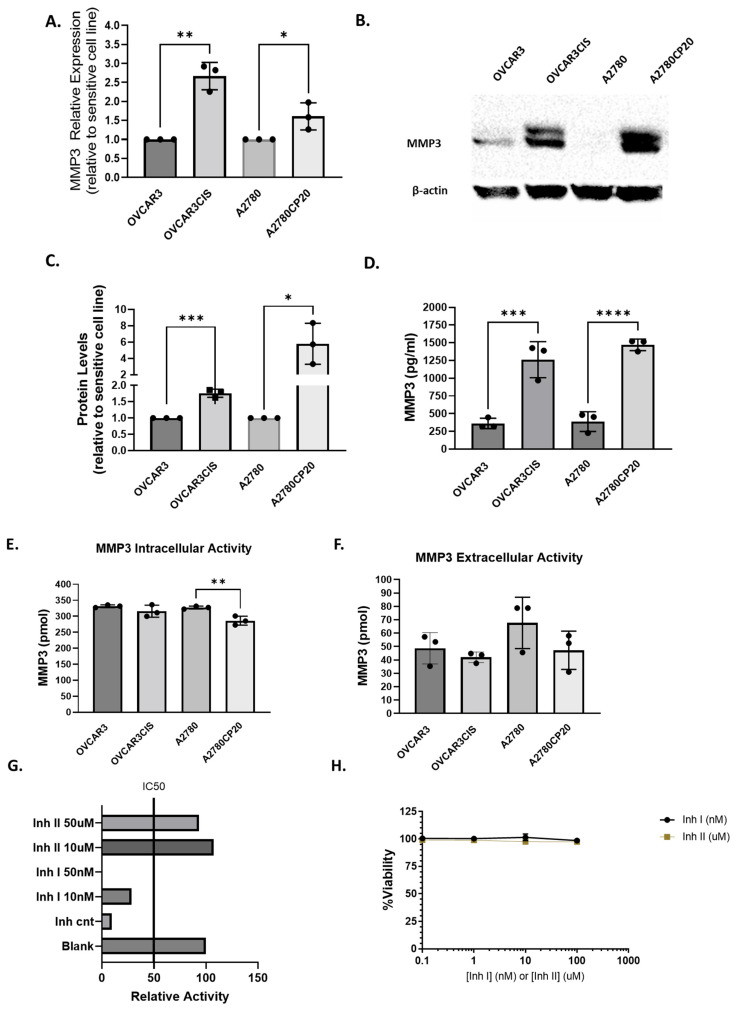
MMP3 is upregulated in cisplatin-resistant ovarian cancer cells; however, its activity does not correlate with cisplatin sensitivity. (**A**–**F**) show the basal MMP3 expression levels measured under untreated conditions. (**A**) RT-qPCR; (**B**) Western blot; (**C**) Densitometric analysis of band intensities and relative values calculated using the intensity of β-actin as a control; (**D**) ELISA of MMP3 relative expression in cisplatin-resistant versus cisplatin-sensitive ovarian cancer cells (OVCAR3 and OVCAR3CIS; A2780 and A2780CP20); (**E**) MMP3 intracellular; and (**F**) extracellular activity in cisplatin-resistant and sensitive cell lines; (**G**) Relative MMP3 activity remaining after MMP3 activity inhibition with Inhibitors I and II in OVCAR3CIS cells; and (**H**) Inhibitor dose−response curve assessed by Alamar blue in OVCAR3CIS cells (* *p* < 0.05, ** *p* < 0.01, *** *p* < 0.001, and **** *p* < 0.0001).

**Figure 2 ijms-26-04012-f002:**
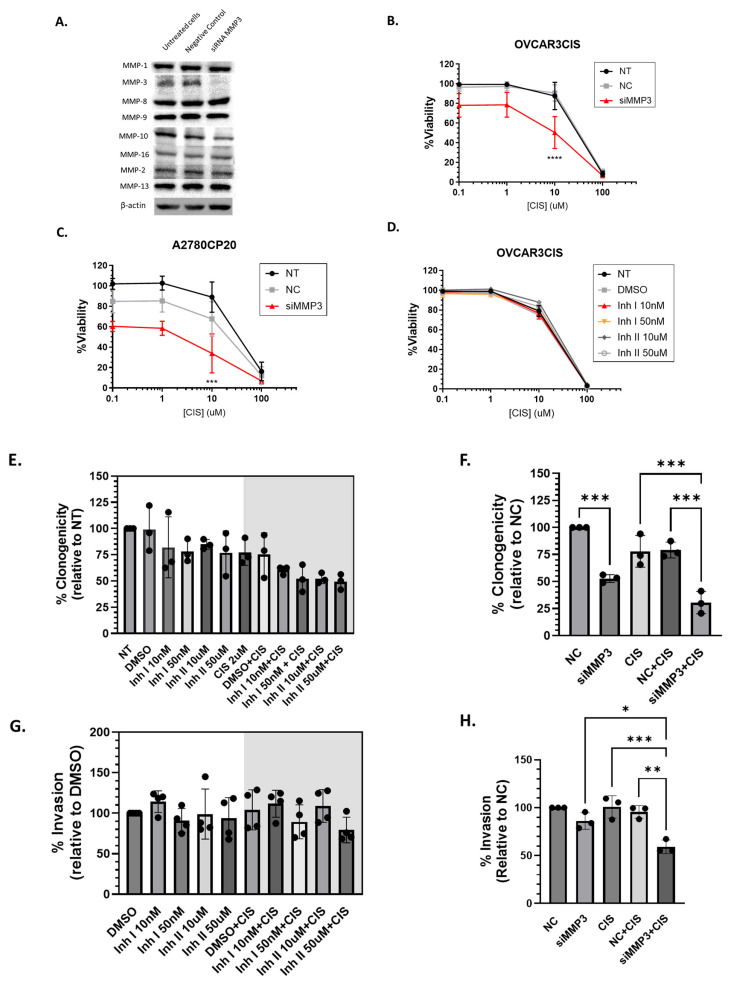
Cisplatin increases MMP3 expression in cisplatin-resistant cell lines, while MMP3 knockdown and catalytic inhibition show divergent effects on cell viability, proliferation, and invasion. (**A**) MMPs expression assessed by Western blot following transfection of MMP3 siRNA (50 nM) in OVCAR3CIS cells; Cell viability in (**B**) OVCAR3CIS and (**C**) A2780CP20 cell viability after MMP3 siRNA transfection, followed by treatment with increasing cisplatin concentrations (*X*-axis). “NT” (not treated) and “NC” (negative control) denote the experimental conditions. (**D**) Cell viability after MMP3 activity inhibition using small-molecule inhibitors, in combination with increasing cisplatin concentrations (*X*-axis; 1–100 μM), increases MMP3 expression in HGSOC, while MMP3 knockdown and catalytic inhibition show divergent effects on cell viability, proliferation, and invasion. Colony formation assay following (**E**) MMP3 activity inhibition with small-molecule inhibitors, (**F**) MMP3 siRNA transfection with and without cisplatin (2 μM) in OVCAR3CIS cells; OVCAR3CIS invasion ability following (**G**) MMP3 small-molecule inhibitors, and (**H**) siRNA transfections with and without cisplatin (10 µM), (**I**) MMP3 expression was assessed by qRT-PCR in OVCAR3CIS treated with cisplatin. All graphs represent mean ± SEM (* *p* < 0.05, ** *p* < 0.01, *** *p* < 0.001, **** *p* < 0.0001, ns: not significant). Experiments were performed at least in triplicate.

**Figure 3 ijms-26-04012-f003:**
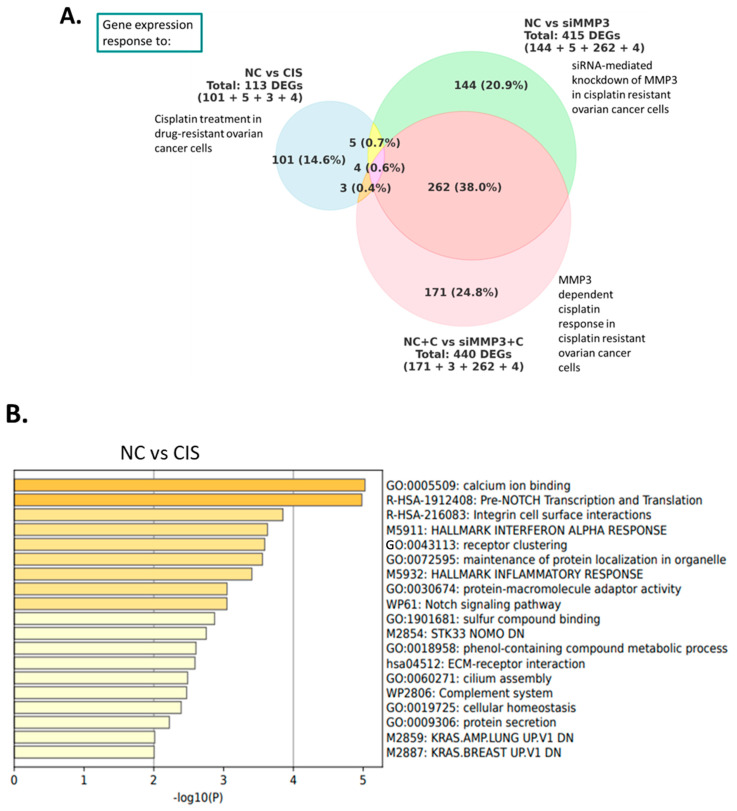
RNA-seq and proteomic analyses. (**A**) Venn diagram of differentially expressed genes under each condition. (**B**) Metascape bar graph showing the top 20 enriched terms among differentially expressed genes (DEGs) in OVCAR3CIS cells, comparing the NC and CIS groups, with a discrete color scale representing statistical significance. (**C**) Metascape bar graph showing the top 20 enriched terms among differentially expressed genes (DEGs) in OVCAR3CIS cells, comparing the NC and siMMP3 groups, with a discrete color scale representing statistical significance. (**D**) Metascape bar graph showing the top 20 enriched terms among differentially expressed genes (DEGs) in OVCAR3CIS cells, comparing the NC + CIS and siMMP3 + CIS groups, with a discrete color scale representing statistical significance. (**E**) Venn diagrams showing MMP3 co-immunoprecipitated proteins in OVCAR3CIS cells and intracellular proteins in non-treated and cisplatin-treated cells. (**F**) Extracellular proteins in non-treated and cisplatin-treated cells. (**G**) Overlapping intracellular and extracellular proteins in cisplatin-treated cells.

**Figure 4 ijms-26-04012-f004:**
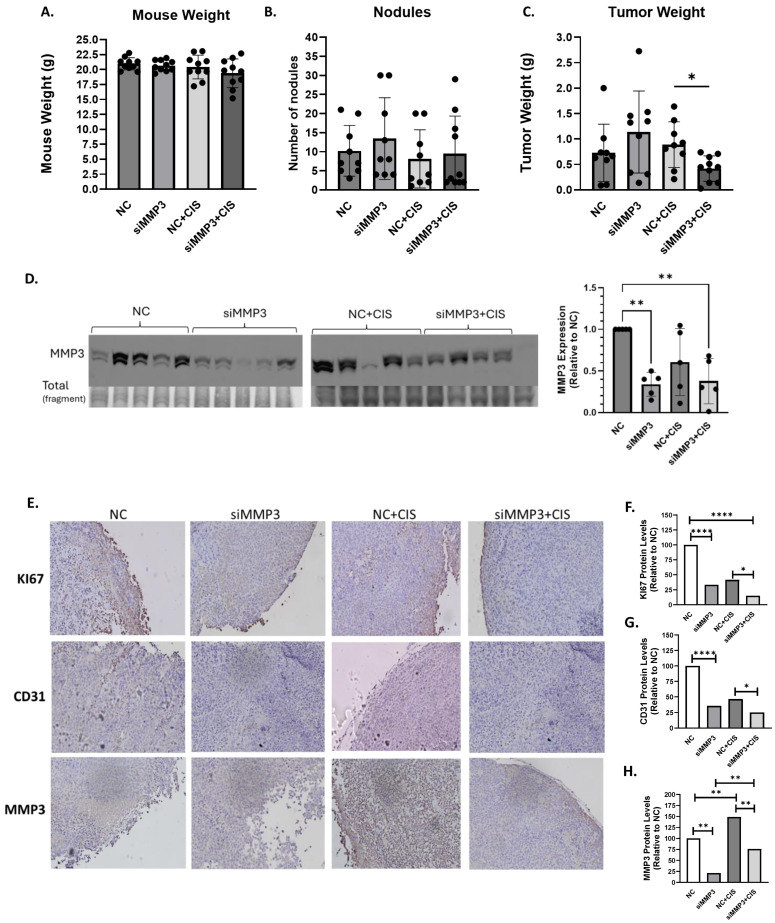
Effects of multiple liposomal-siRNA-MMP3 injections and immunohistochemical analysis of MMP3 targeting in an ovarian cancer mouse model. (**A**) mouse weight, (**B**) number of nodules, (**C**) tumor weight in all conditions, (**D**) MMP3 expression in mice tumors was assessed by Western blot following multiple injections of liposomal-siRNA-MMP3 alone or in combination with cisplatin and densitometric analysis of band intensities and relative values calculated using the intensity of the shown total protein fragment as control, (**E**) tissue sections from OVCAR3CIS tumors immunostained for the detection of proliferation by measuring Ki-67, CD31 for blood vessel formation and MMP3. Images captured using 20× magnification. Protein levels of (**F**) Ki-67 (proliferation), (**G**) CD31 (blood vessel formation), and (**H**) MMP3 in OVCAR3CIS tumors were quantified across five fields for each tissue sample and using four mice per group (* *p* < 0.05, ** *p* < 0.01, **** *p* < 0.0001).

**Figure 5 ijms-26-04012-f005:**
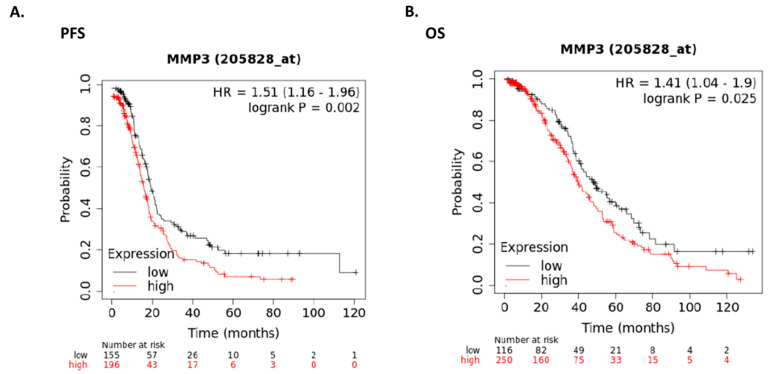
Expression of MMP3 in patients with human ovarian cancer. Kaplan-Meier plots showing (**A**) progression-free survival (PFS) analysis and (**B**) overall survival (OS) of patients with HGSOC with low and high levels of MMP3.

**Table 1 ijms-26-04012-t001:** Top 20 differentially expressed genes (DEGs) associated with siRNA-mediated MMP3 knockdown, MMP3-dependent cisplatin response, and siRNA-mediated MMP3 knockdown plus cisplatin compared with NC-siRNA plus cisplatin in OVCAR3CIS cells. The list is ranked by the adjusted *p*-value (*p*-adj).

Gene Name	ID	log2FoldChange	*p*-adj
*siRNA-mediated MMP3 knockdown*			
SCAMP5	ENSG00000198794	1.060988	1.76 × 10^−51^
NEFH	ENSG00000100285	1.139433	2.61 × 10^−40^
BTG2	ENSG00000159388	−1.09713	4.09 × 10^−26^
VASP	ENSG00000125753	−1.05933	5.52 × 10^−25^
ITGA5	ENSG00000161638	−1.00827	5.68 × 10^−24^
WWTR1	ENSG00000018408	−1.03085	2.2 × 10^−22^
PHF21B	ENSG00000056487	1.042219	9.14 × 10^−22^
RPS6KA4	ENSG00000162302	−1.02319	1.39 × 10^−21^
DLX2	ENSG00000115844	−1.22783	1.52 × 10^−20^
RELL2	ENSG00000164620	1.035967	9.42 × 10^−16^
KCNQ2	ENSG00000075043	1.043291	3.8 × 10^−15^
SERPINB9	ENSG00000170542	−1.02407	1.95 × 10^−11^
SNCB	ENSG00000074317	1.281406	5.91 × 10^−11^
SVOP	ENSG00000166111	4.904506	4.58 × 10^−10^
MPC1	ENSG00000060762	1.037979	6.96 × 10^−10^
GPR3	ENSG00000181773	−1.12902	7.66 × 10^−10^
ERO1A	ENSG00000197930	−1.01424	8.17 × 10^−10^
C3orf14	ENSG00000114405	1.073282	9.45 × 10^−10^
LMLN	ENSG00000185621	−1.0919	2.97 × 10^−9^
FGF7P3	ENSG00000204837	1.163488	9.88 × 10^−9^
*MMP3-dependent cisplatin response*			
CHGB	ENSG00000089199	3.603976	9.84 × 10^−154^
CHRNB2	ENSG00000160716	4.26102	7.07 × 10^−137^
SCG3	ENSG00000104112	5.04094	2.42 × 10^−130^
AP3B2	ENSG00000103723	2.221219	4.52 × 10^−122^
SYP	ENSG00000102003	2.309564	1.06 × 10^−121^
STMN3	ENSG00000197457	2.493578	1.14 × 10^−110^
RUNDC3A	ENSG00000108309	3.001584	2.17 × 10^−102^
ACTL6B	ENSG00000077080	4.645848	1.06 × 10^−96^
XKR7	ENSG00000260903	4.393692	3.97 × 10^−85^
MAPK8IP2	ENSG00000008735	2.266975	1.26 × 10^−72^
CPLX1	ENSG00000168993	3.580086	2.59 × 10^−71^
NFE2L2	ENSG00000116044	−1.23975	2.19 × 10^−59^
UNC79	ENSG00000133958	2.025565	1.71 × 10^−57^
MAPK8IP1	ENSG00000121653	1.378141	3.56 × 10^−55^
PAX5	ENSG00000196092	3.248805	3.04 × 10^−52^
CHGA	ENSG00000100604	3.290491	2.07 × 10^−49^
AC005696.4	ENSG00000277200	3.286964	1.39 × 10^−48^
FAM57B	ENSG00000149926	2.656496	3.01 × 10^−48^
INIP	ENSG00000148153	−1.25455	9.29 × 10^−47^
DPY19L1	ENSG00000173852	−1.36383	2.2 × 10^−44^
*siRNA-MMP3 plus cisplatin vs. NC-siRNA plus cis-platin*			
CDC25A	ENSG00000164045	−1.0636	4.63 × 10^−44^
ANKRD1	ENSG00000148677	−1.0090	3.96 × 10^−39^
ACTR2	ENSG00000138071	−1.1531	3.37 × 10^−37^
GFPT1	ENSG00000198380	−1.0276	1.58 × 10^−34^
KCNC3	ENSG00000131398	1.1819	9.43 × 10^−33^
CARNMT1	ENSG00000156017	−1.0228	6.55 × 10^−32^
PLEKHB2	ENSG00000115762	−1.0282	2.73 × 10^−31^
PODXL2	ENSG00000114631	1.0052	1.41 × 10^−27^
LIMA1	ENSG00000050405	−1.0028	5.53 × 10^−27^
REEP5	ENSG00000129625	−1.0534	3.17 × 10^−26^
UEVLD	ENSG00000151116	−1.0062	8.64 × 10^−25^
SBK1	ENSG00000188322	1.1215	9.09 × 10^−22^
POLR3G	ENSG00000113356	−1.0585	7.21 × 10^−21^
ADAMTSL4	ENSG00000143382	1.0051	7.96 × 10^−19^
MOSPD3	ENSG00000106330	1.0839	3.99 × 10^−18^
TRIP11	ENSG00000100815	−1.0167	3.2 × 10^−17^
TIMM10	ENSG00000134809	−1.1039	1.03 × 10^−16^
PDCD4	ENSG00000150593	−1.0309	1.07 × 10^−16^
F11R	ENSG00000158769	−1.0167	1.46 × 10^−16^
CNNM1	ENSG00000119946	1.0436	1.82 × 10^−15^

**Table 2 ijms-26-04012-t002:** Intracellular proteins Co-IP with MMP3 in untreated and cisplatin-treated OVCAR3CIS cells.

Gene Symbol	Description	MW [kDa]	calc. pI	ENSEMBL Gene ID
*unique intracellular proteins in cisplatin-treated cells*				
ALB	Albumin	69.3	6.28	ENSG00000163631
C18orf63	Uncharacterized protein C18orf63	77.2	9.8	ENSG00000206043
CCDC22	Coiled-coil domain-containing protein 22	70.7	6.74	ENSG00000101997
GAPDH	Glyceraldehyde-3-phosphate dehydrogenase	36	8.46	ENSG00000111640
H2BC21	Histone H2B type 2-E	13.9	10.32	ENSG00000184678
HSPA9	Stress-70 protein, mitochondrial	73.6	6.16	ENSG00000113013
KCTD3	BTB/POZ domain-containing protein KCTD3	88.9	7.03	ENSG00000136636
MYL12B	Myosin regulatory light chain 12B	19.8	4.84	ENSG00000118680
RPL27A	Large ribosomal subunit protein uL15	16.6	11	ENSG00000166441
RPL36AL	Ribosomal protein eL42-like	12.5	10.65	ENSG00000165502
RPS15	Small ribosomal subunit protein uS19	17	10.39	ENSG00000115268
RPS23	Small ribosomal subunit protein uS12	15.8	10.49	ENSG00000186468
RPS29	Small ribosomal subunit protein uS14	6.7	10.13	ENSG00000213741
RPS4Y2	Small ribosomal subunit protein eS4, Y isoform 2	29.3	10.08	ENSG00000280969
RPS6	Small ribosomal subunit protein eS6	28.7	10.84	ENSG00000137154
SRXN1	Sulfiredoxin-1	14.3	8.19	ENSG00000271303
*Common proteins in cisplatin-treated and cisplatin-untreated cells*				
ACTB	Actin, cytoplasmic 1	41.7	5.48	ENSG00000075624
ATXN2	Ataxin-2	140.2	9.57	ENSG00000204842
BANF1	Barrier-to-autointegration factor	10.1	6.09	ENSG00000175334
CLTC	Clathrin heavy chain 1	191.5	5.69	ENSG00000141367
GRN	Progranulin	63.5	6.83	ENSG00000030582
H2AC20	Histone H2A type 2-C	14	10.9	ENSG00000184260
H4C1; H4C11; H4C12; H4C13; H4C14; H4C15; H4C16; H4C2; H4C3; H4C4; H4C5; H4C6; H4C8; H4C9	Histone H4	11.4	11.36	ENSG00000158406; ENSG00000197061; ENSG00000197238; ENSG00000197837; ENSG00000270276; ENSG00000270882; ENSG00000273542; ENSG00000274618; ENSG00000275126; ENSG00000276180; ENSG00000276966; ENSG00000277157; ENSG00000278637; ENSG00000278705
IGHG1	Isoform 1 of Immunoglobulin heavy constant gamma 1	36.1	8.19	ENSG00000211896
MYH10	Myosin-10	228.9	5.54	ENSG00000133026
MYH9	Myosin-9	226.4	5.6	ENSG00000100345
MYL6	Myosin light polypeptide 6	16.9	4.65	ENSG00000092841
RPL13	Large ribosomal subunit protein eL13	24.2	11.65	ENSG00000167526
RPL15	Large ribosomal subunit protein eL15	24.1	11.62	ENSG00000174748
RPL35	Large ribosomal subunit protein uL29	14.5	11.05	ENSG00000136942
RPL39	Large ribosomal subunit protein eL39	6.4	12.56	ENSG00000198918
RPS11	Small ribosomal subunit protein uS17	18.4	10.3	ENSG00000142534
RPS13	Small ribosomal subunit protein uS15	17.2	10.54	ENSG00000110700
RPS14	Small ribosomal subunit protein uS11	16.3	10.05	ENSG00000164587
RPS18	Small ribosomal subunit protein uS13	17.7	10.99	ENSG00000096150; ENSG00000223367; ENSG00000226225; ENSG00000231500; ENSG00000235650
RPS24	Small ribosomal subunit protein eS24	15.4	10.78	ENSG00000138326
RPS25	Small ribosomal subunit protein eS25	13.7	10.11	ENSG00000118181; ENSG00000280831
RPS3	Small ribosomal subunit protein uS3	26.7	9.66	ENSG00000149273
*unique proteins in cisplatin untreated cells*				
FLG	Filaggrin	434.9	9.25	ENSG00000143631
H2AX	Histone H2AX	15.1	10.74	ENSG00000188486
H3-3A; H3-3B	Histone H3.3	15.3	11.27	ENSG00000132475; ENSG00000163041
HMGA2	High mobility group protein HMGI-C	11.8	10.62	ENSG00000149948
WDFY1	WD repeat and FYVE domain-containing protein 1	46.3	7.33	ENSG00000085449

MW [kDa] refers to the calculated molecular weight of the protein in kilodaltons, while calc. pI represents the theoretical isoelectric point.

**Table 3 ijms-26-04012-t003:** Extracellular proteins Co-IP with MMP3 in untreated and cisplatin-treated OVCAR3CIS cells.

Gene Symbol	Description	MW [kDa]	calc. pI	ENSEMBL Gene ID
*unique intracellular proteins in cisplatin-treated cells*				
ACTB	Actin, cytoplasmic 1	41.7	5.48	ENSG00000075624
CCDC22	Coiled-coil domain-containing protein 22	70.7	6.74	ENSG00000101997
HNRNPAB	Heterogeneous nuclear ribonucleoprotein A/B	36.2	8.21	ENSG00000197451
ALPK2	Alpha-protein kinase 2	236.9	5.24	ENSG00000198796
RPS25	Small ribosomal subunit protein eS25	13.7	10.11	ENSG00000118181; ENSG00000280831
KIAA0232	Uncharacterized protein KIAA0232	154.7	4.78	ENSG00000170871
IGHG4	Immunoglobulin heavy constant gamma 4	43.8	6.24	ENSG00000211892; ENSG00000277016
H4C1; H4C11; H4C12; H4C13; H4C14; H4C15; H4C16; H4C2; H4C3; H4C4; H4C5; H4C6; H4C8; H4C9	Histone H4	11.4	11.36	ENSG00000158406; ENSG00000197061; ENSG00000197238; ENSG00000197837; ENSG00000270276; ENSG00000270882; ENSG00000273542; ENSG00000274618; ENSG00000275126; ENSG00000276180; ENSG00000276966; ENSG00000277157; ENSG00000278637; ENSG00000278705
POTEE	POTE ankyrin domain family member E	121.3	6.2	ENSG00000188219
RPL26	Large ribosomal subunit protein uL24	17.2	10.55	ENSG00000161970
DST	Dystonin	860.1	5.25	ENSG00000151914
NPAP1	Nuclear pore-associated protein 1	120.9	8.69	ENSG00000185823
ATP10B	Phospholipid-transporting ATPase VB	165.3	6.89	ENSG00000118322
*Common proteins in cisplatin-treated and cisplatin-untreated cells*				
IGHG1	Isoform 1 of Immunoglobulin heavy constant gamma 1	36.1	8.19	ENSG00000211896
ATXN2	Ataxin-2	140.2	9.57	ENSG00000204842
IGLC1	Immunoglobulin lambda-1 light chain	22.8	6.76	ENSG00000211675
LAMA5	Laminin subunit alpha-5	399.5	7.02	ENSG00000130702
ALB	Albumin	69.3	6.28	ENSG00000163631
FLG	Filaggrin	434.9	9.25	ENSG00000143631
C1R	Complement C1r subcomponent	80.1	6.21	ENSG00000159403
HSPA7	Putative heat shock 70 kDa protein 7	40.2	7.87	ENSG00000225217
*unique proteins in cisplatin untreated cells*				
AFF1	AF4/FMR2 family member 1	131.3	9.2	ENSG00000172493
C18orf63	Uncharacterized protein C18orf63	77.2	9.8	ENSG00000206043
CCDC25	Coiled-coil domain-containing protein 25	24.5	6.8	ENSG00000147419
DEFA1; DEFA1B	Neutrophil defensin 1	10.2	6.99	ENSG00000206047; ENSG00000240247; ENSG00000284983; ENSG00000285176
DSC1	Desmocollin-1	99.9	5.43	ENSG00000134765
FAM228B	Protein FAM228B	38	8.73	ENSG00000219626
GRN	Progranulin	63.5	6.83	ENSG00000030582
HBE1	Hemoglobin subunit epsilon	16.2	8.63	ENSG00000213931
IGLV2-11	Immunoglobulin lambda variable 2–11	12.6	7.24	ENSG00000211668
LILRB3	Leukocyte immunoglobulin-like receptor subfamily B member 3	69.3	6.93	ENSG00000204577; ENSG00000274587; ENSG00000275019
MGAT5B	Alpha-1,6-mannosylglycoprotein 6-beta-N-acetylglucosaminyltransferase B	89.5	8.35	ENSG00000167889
MORC4	MORC family CW-type zinc finger protein 4	106.3	7.46	ENSG00000133131
PABPN1L	Isoform 2 of Embryonic polyadenylate-binding protein 2	27.4	4.72	
POTEF	POTE ankyrin domain family member F	121.4	6.2	ENSG00000196604
RECQL4	ATP-dependent DNA helicase Q4	133	8.09	ENSG00000160957
RP1L1	Retinitis pigmentosa 1-like 1 protein	252.1	4.45	ENSG00000183638
SMARCD2	SWI/SNF-related matrix-associated actin-dependent regulator of chromatin subfamily D member 2	58.9	9.64	ENSG00000108604
SMIM35	Small integral membrane protein 35	9.4	5.87	ENSG00000255274
SORBS1	Sorbin and SH3 domain-containing protein 1	142.4	6.84	ENSG00000095637
SPTAN1	Spectrin alpha chain, non-erythrocytic 1	284.4	5.35	ENSG00000197694
TRIM41	E3 ubiquitin-protein ligase TRIM41	71.6	5.06	ENSG00000146063

MW [kDa] refers to the calculated molecular weight of the protein in kilodaltons, while calc. pI represents the theoretical isoelectric point.

## Data Availability

All data generated or analyzed during this study are included in this published article (and its Appendix A). The RNA-seq data that support the findings of this study were deposited in the NCBI’s Gene Expression Omnibus and are accessible through https://www.ncbi.nlm.nih.gov/geo/query/acc.cgi?acc=GSE275051 GEO Series accession number [GSE275051].

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
