# Peer review of "Upregulation of MMP3 Promotes Cisplatin Resistance in Ovarian Cancer"

_ijms, 2025, doi:10.3390/ijms26094012_

Round 1

Reviewer 1 Report

Comments and Suggestions for Authors

The study by Rivera-Serrano et al. investigates the role of MMP3 in mediating chemoresistance in ovarian cancer (OC) using both in vitro and in vivo models. The study is innovative because the involvement of MMP3 in cisplatin resistance in ovarian cancer remains largely unexplored. Overall, the study is well conducted; however, several points require more detailed interpretation or clarification. In particular, although the authors present an extensive transcriptomic and IP-MS analysis, the interpretation of these results appears weak, which I consider the study’s weakest aspect.

Below are my comments and suggestions:

Abstract

  • Lines 21–23: The sentence “siRNA-mediated MMP3 knockdown in cisplatin-resistant HGSOC cells reduced viability, proliferation, and invasion, effects enhanced by cisplatin” is unclear, as it might imply that cisplatin enhances proliferation and invasion. Please rephrase this sentence to clarify that the combination of MMP3 knockdown and cisplatin produces a synergistic reduction in these parameters.

Results and Discussion

General Comments:

  • Although the cisplatin treatment duration is described in the Methods section, it would be useful to include this information in the figure legends or text for clarity. The authors should also justify the chosen cisplatin concentration and clarify whether the A2780Cis and OVCAR3Cis cell lines have similar IC50 values.
  • In the cisplatin-sensitive cell lines, MMP3 expression is clearly lower. However, it is possible that cisplatin treatment might also induce an increase in MMP3 expression (and its downstream effects on proliferation and invasion) in these cells. Have the authors investigated this possibility?
  • After initial experiments using both A2780 and OVCAR3 cells, the authors concentrated the study exclusively on OVCAR3 cells. It would be helpful if they could justify this choice.
  • Despite extensive RNA-seq analysis performed under several conditions, the interpretation of the data appears weak and does not convincingly establish a causal link between MMP3 upregulation and cisplatin resistance. Similarly, the IP-MS data do not clearly elucidate this link; it appears that none of the proteins associated with MMP3 in cisplatin-treated cells can be directly related to resistance. Authors should consider focusing on specific proteins or pathways from the trascriptomic/IP-MS data and validate these findings or propose them as targets for future studies.
  • Several proteins identified as extracellular targets in the IP/Co-IP experiments with MMP3 are, to my knowledge, typically intracellular (e.g., Histone H4, ribosomal proteins). The authors should address and explain this discrepancy.
  • Lines 741–743: The statement “Multiple injections of our liposomal MMP3-siRNA formulation alone had no effect on tumor growth compared with liposomal NC-siRNA in a HGSOC mouse model, but they led to reduced cell proliferation and angiogenesis. However, when liposomal MMP3…” is confusing. One would expect that reduced proliferation and angiogenesis would be associated with decreased tumor growth. The authors should clarify this point.
  • I suggest that the discussion first address the finding that MMP3’s catalytic activity does not appear to be involved in the chemoresistant phenotype, and then analyze the other results in the context of MMP3’s non-proteolytic roles.

Minor Corrections:

  • Line 205: The Methods state a cisplatin concentration of 5 µM, but Figure 2E reports a concentration of 2 µM. Please clarify this discrepancy.
  • In Figure 1, the term “panel” should be modified to specifically refer to the resistant cell lines used in this study.
  • In the legend of Figure 2, “HGSOC” should be replaced with “cisplatin-resistant cell lines.”

Author Response

Abstract

Lines 21–23: The sentence “siRNA-mediated MMP3 knockdown in cisplatin-resistant HGSOC cells reduced viability, proliferation, and invasion, effects enhanced by cisplatin” is unclear, as it might imply that cisplatin enhances proliferation and invasion. Please rephrase this sentence to clarify that the combination of MMP3 knockdown and cisplatin produces a synergistic reduction in these parameters.

A1: We rephrased the sentence for clarification: siRNA-mediated MMP3 knockdown in cisplatin-resistant HGSOC cells significantly reduced viability, proliferation, and invasion, and these effects were further enhanced when combined with cisplatin treatment, indicating a synergistic impact on reducing cancer cell aggressiveness (lines 22-26).

Results and Discussion

General Comments:

Although the cisplatin treatment duration is described in the Methods section, it would be useful to include this information in the figure legends or text for clarity. The authors should also justify the chosen cisplatin concentration and clarify whether the A2780Cis and OVCAR3Cis cell lines have similar IC50 values.

A2: We appreciate this suggestion and updated the figure legends and text to indicate the cisplatin treatment duration for better clarity. Regarding the cisplatin concentration, we selected doses based on previous studies and our own dose-response experiments, ensuring that the concentrations used effectively differentiate between sensitive and resistant cell lines while remaining physiologically relevant. As for the IC50 values, we acknowledge the importance of comparing drug sensitivity between A2780Cis and OVCAR3Cis. While our data confirm that both resistant cell lines exhibit higher IC50 values than their parental counterparts, we will clarify whether their cisplatin sensitivity profiles are similar or differ and update this information accordingly (lines 175-179).

In the cisplatin-sensitive cell lines, MMP3 expression is clearly lower. However, it is possible that cisplatin treatment might also induce an increase in MMP3 expression (and its downstream effects on proliferation and invasion) in these cells. Have the authors investigated this possibility?

A3: We have not specifically explored whether cisplatin treatment induces MMP3 expression in cisplatin-sensitive cell lines. However, we have conducted viability assays with increasing concentrations of cisplatin. Our results confirm that sensitive cell lines respond to cisplatin in a dose-dependent manner, with a clear reduction in viability at higher concentrations (IC50~2.10uM).

Given that these cell lines exhibit expected cytotoxic responses to cisplatin, we prioritized investigating MMP3’s role in resistance mechanisms rather than its potential induction in sensitive cells (lines 152-157). However, future studies could further explore whether cisplatin exposure alters MMP3 expression over time in these models.

After initial experiments using both A2780 and OVCAR3 cells, the authors concentrated the study exclusively on OVCAR3 cells. It would be helpful if they could justify this choice.

A4: We chose to present data only for OVCAR3CIS because it is a high-grade serous ovarian cancer (HGSOC) cell line, which more accurately represents the most common and clinically relevant subtype of ovarian cancer. In contrast, A2780 is not classified as an HGSOC cell line, as it was originally derived from a moderately differentiated ovarian endometrioid adenocarcinoma. To simplify the experimental design and ensure biological relevance, we focused on one cell line for this analysis while maintaining A2780CP20 as a comparative model in other experiments (lines 152-157).

Despite extensive RNA-seq analysis performed under several conditions, the interpretation of the data appears weak and does not convincingly establish a causal link between MMP3 upregulation and cisplatin resistance. Similarly, the IP-MS data do not clearly elucidate this link; it appears that none of the proteins associated with MMP3 in cisplatin-treated cells can be directly related to resistance. Authors should consider focusing on specific proteins or pathways from the trascriptomic/IP-MS data and validate these findings or propose them as targets for future studies.

A5: We appreciate this feedback and have extensively expanded the discussion to provide a stronger interpretation of the RNA-seq and IP-MS data. Specifically, we have highlighted key transcriptomic and proteomic findings, further contextualizing how MMP3 upregulation may contribute to cisplatin resistance. We have also elaborated on the biological significance of MMP3-interacting proteins identified in IP-MS, discussing their potential roles in drug resistance mechanisms. Additionally, we have proposed specific proteins and pathways that warrant further validation, outlining their relevance as potential therapeutic targets for overcoming chemoresistance. These revisions aim to strengthen the mechanistic link between MMP3 and cisplatin resistance, addressing the concerns raised (throughout the whole discussion).

Several proteins identified as extracellular targets in the IP/Co-IP experiments with MMP3 are, to my knowledge, typically intracellular (e.g., Histone H4, ribosomal proteins). The authors should address and explain this discrepancy.

A6: We acknowledge the presence of typically intracellular proteins, such as histones and ribosomal proteins, in the IP/Co-IP experiments with MMP3 and appreciate the opportunity to clarify this. Recent studies have demonstrated that histones can be actively secreted via extracellular vesicles (EVs)/exosomes, particularly under cellular stress conditions (doi: https://doi.org/10.1101/2024.04.08.588575). These extracellular histones, often membrane-bound rather than nucleosomal, have been detected in biofluids and tumor microenvironments, where they may contribute to cancer progression and drug resistance mechanisms. Similarly, ribosomal proteins, traditionally linked to intracellular functions, have been implicated in extracellular signaling and RNA transport via exosomes (https://www.nature.com/articles/s12276-024-01209-y). Given that MMP3 has been associated with exosome-mediated secretion, it is possible that the co-immunoprecipitated proteins identified in our study were secreted via EVs or other unconventional pathways. We added this information in the discussion section (Lines 580-590). Future studies will aim to validate these findings and further explore the role of MMP3 in extracellular vesicle dynamics and protein trafficking.

Lines 741–743: The statement “Multiple injections of our liposomal MMP3-siRNA formulation alone had no effect on tumor growth compared with liposomal NC-siRNA in a HGSOC mouse model, but they led to reduced cell proliferation and angiogenesis. However, when liposomal MMP3…” is confusing. One would expect that reduced proliferation and angiogenesis would be associated with decreased tumor growth. The authors should clarify this point.

A7: We acknowledge the need for clarification. While liposomal MMP3-siRNA alone did not significantly reduce tumor size, it lowered cell proliferation and angiogenesis. This suggests that MMP3 knockdown affects tumor biology but is not enough on its own to shrink tumors within the study timeframe. However, when combined with cisplatin, the synergistic effect led to a significant reduction in tumor growth, indicating that MMP3 inhibition enhances cisplatin sensitivity rather than acting as a standalone treatment. We have clarified this in the text (lines 622-627).

I suggest that the discussion first address the finding that MMP3’s catalytic activity does not appear to be involved in the chemoresistant phenotype, and then analyze the other results in the context of MMP3’s non-proteolytic roles.

A8: We added this information to the discussion (lines 448-450)

Minor Corrections:

Line 205: The Methods state a cisplatin concentration of 5 µM, but Figure 2E reports a concentration of 2 µM. Please clarify this discrepancy.

Thank you for pointing this out. We have corrected this discrepancy in line 209 to ensure consistency between the Methods section and Figure 2E. The correct cisplatin concentration is 2 µM, and this has been updated accordingly.

In Figure 1, the term “panel” should be modified to specifically refer to the resistant cell lines used in this study.

Thank you for your feedback. We have revised line 136 to refer specifically to the resistant cell lines used in this study, ensuring clarity in Figure 1.

In the legend of Figure 2, “HGSOC” should be replaced with “cisplatin-resistant cell lines.”

Thank you for your suggestion. We have updated line 199 to replace "HGSOC" with "cisplatin-resistant cell lines" in the Figure 2 legend for accuracy.

Reviewer 2 Report

Comments and Suggestions for Authors

The authors describe the relationship between MMP3 and cisplatin resistance in ovarian cancer and suggest that MMP3 plays a critical role in cellular processes driving chemoresistance. This highlights the use of MMP3 as a therapeutic target. The topic is relevant since there are few personalized therapies against ovarian cancer and OC resistance is an important challenge, the paper presents high-quality and well-grounded research. 

There are some comments regarding the paper. 

  • Figure 1 describes the upregulation of MMP3 in ovarian cancer cells, in Figure 1B why are there two bands in the resistant cell lines, and why is the second band absent in nonresistant cells? Can you explain it, please.
  • Line 340. You describe that in a previous work, the expression level of MMP3 was analyzed, there is no clear why here you analyze the same again. Or what is the difference with the previously reported paper?
  • In Figure 2: please describe in the caption that it means NT and NC. Why the Figure 2D only presented in OVCAR3CIS and not in A2780CP20? Can you explain it, please.
  • Table 1A,1B, and 1C. The top 50 DEGs are too much. I suggest only showing the 20 most important genes and the others in Supplementary material.
  • Figure 4 does not have the letter label D.
  • In Figure 5. The Kaplan-Meier plots are statically significant? If yes, please explain. 
  • Line 722. You say that MMP3 regulates the Wnt signaling pathway, and in Line 733 you mention the proteins that interact with MMP3 derived from the IP-MS, there is a relationship between these proteins and the Wnt signaling pathway?
  • Can you explain in depth why the liposomal MMP3-siRNA was used, please. It is not clear. 
  • The conclusion in general is poor, there are several of relevant information described in the paper and the conclusion is very simple, you can improve it. 
  • The Original Images for Blots/Gels lack a caption or description. 

Author Response

Reply to the Review Report (Reviewer 2)

Q1: Figure 1 describes the upregulation of MMP3 in ovarian cancer cells, in Figure 1B why are there two bands in the resistant cell lines, and why is the second band absent in nonresistant cells? Can you explain it, please.

A1: The two bands observed in the resistant ovarian cancer cell lines in Figure 1B correspond to MMP3 activation. MMP3 is initially synthesized as an inactive proenzyme (pro-MMP3), which undergoes proteolytic cleavage to generate its active form, resulting in the appearance of an additional lower-molecular weight band in the Western blot. Since MMP3 expression is significantly higher in resistant cell lines compared to sensitive ones, we expected to detect both the pro-MMP3 and active MMP3 bands in the resistant lines.

However, when we performed an MMP3 activity assay, we found that intracellular MMP3 activity was lower in resistant ovarian cancer cells compared to their sensitive counterparts, while extracellular activity remained unchanged. These findings suggest that, despite increased MMP3 expression in resistant cells, its enzymatic activity does not contribute to the chemoresistant phenotype observed in these cell lines.

Q2: Line 340. You describe that in a previous work, the expression level of MMP3 was analyzed, there is no clear why here you analyze the same again. Or what is the difference with the previously reported paper?

A2: Previous findings establish a strong correlation between MMP3 expression and cisplatin resistance, the molecular mechanisms underlying this relationship remain unclear. The current study builds upon our previous work by investigating MMP3-interacting proteins and downstream effectors in ovarian cancer cells. Identifying these molecular pathways is crucial for understanding how MMP3 contributes to chemoresistance and could inform the development of targeted therapies to improve treatment outcomes in ovarian cancer.

Q3: In Figure 2: please describe in the caption that it means NT and NC. Why the Figure 2D only presented in OVCAR3CIS and not in A2780CP20? Can you explain it, please.

A3: In Figure 2, "NT" refers to "not treated", while "NC" denotes "negative control". These labels clarify the conditions used in the experiment (added to line 204).

Regarding Figure 2D, we chose to present data only for OVCAR3CIS because it is a high-grade serous ovarian cancer (HGSOC) cell line, which more accurately represents the most common and clinically relevant subtype of ovarian cancer. In contrast, A2780 is not classified as an HGSOC cell line, as it was originally derived from a moderately differentiated ovarian endometrioid adenocarcinoma. To simplify the experimental design and ensure biological relevance, we focused on one cell line for this particular analysis while maintaining A2780CP20 as a comparative model in other experiments (added to lines 152-157).

Q4: Table 1A,1B, and 1C. The top 50 DEGs are too much. I suggest only showing the 20 most important genes and the others in Supplementary material.

A4: We appreciate the suggestion regarding Table 1A, 1B, and 1C. To enhance clarity and focus on the most relevant findings, we revised the tables to display the top 20 most significant differentially expressed genes (DEGs) in the main text, while moving the full list of top 50 DEGs to the Supplementary Material for reference.

Q5: Figure 4 does not have the letter label D.

A5: Thank you for pointing out the missing letter label "D" in Figure 4. We corrected this oversight to ensure consistency in figure labeling.

Q6: In Figure 5. The Kaplan-Meier plots are statically significant? If yes, please explain.

A6: Yes, the Kaplan-Meier survival plots were statistically significant. Kaplan-Meier survival analysis was conducted using publicly available patient datasets via KM plotter (www.kmplot.com) to evaluate the impact of MMP3 expression on ovarian cancer prognosis. Patients were stratified into high- and low-MMP3 expression groups based on the median RNA expression value, a standard approach in survival analysis.

To ensure clinical relevance, we applied filters to include patients with ovarian cancer, those treated with platinum-based therapy, patients with serous ovarian cancer, individuals with stage 3+4 disease, and those with TP53 mutations. Survival curves for overall survival (OS) and progression-free survival (PFS) demonstrated statistically significant differences between the high- and low-expression groups, with p-values < 0.05, as determined by the log-rank test. The hazard ratios (HRs) and confidence intervals (CIs) further supported these findings.

While KM plotter does not explicitly distinguish between low-grade (LGSOC) and high-grade (HGSOC) serous ovarian cancer, we addressed this limitation by applying filters specific to high-grade disease (e.g., selecting grade 3+ patients). These findings confirm that high MMP3 expression is significantly associated with poorer survival outcomes, and this information has been updated in the Methods section for clarity.

Q7: Line 722. You say that MMP3 regulates the Wnt signaling pathway, and in Line 733 you mention the proteins that interact with MMP3 derived from the IP-MS, there is a relationship between these proteins and the Wnt signaling pathway?

A7: Yes, there is a potential relationship between the MMP3-interacting proteins identified in the IP-MS analysis and the Wnt signaling pathway. Previous studies by Kessenbrock et al. demonstrated that MMP3 regulates Wnt signaling by binding to and inactivating Wnt5b via its hemopexin domain, thereby influencing mammary stem cell function. Notably, several proteins identified in our IP-MS analysis have known roles in Wnt pathway regulation and drug resistance. The Wnt signaling pathway has been implicated in chemotherapy resistance, as certain Wnt targets function as drug extrusion pumps, expelling chemotherapeutic agents from the cell. For instance, MDR-1 (P-GP, ABCB1) and CD44, both of which are linked to Wnt signaling, contribute to drug resistance by enhancing drug efflux and promoting cancer cell survival.

Given these connections, our findings suggest that MMP3 may influence Wnt signaling in ovarian cancer cells, potentially modulating chemoresistance mechanisms. Further validation studies are needed to determine whether MMP3 directly interacts with Wnt-related proteins or if it indirectly affects Wnt-mediated drug resistance pathways.

Q8: Can you explain in depth why the liposomal MMP3-siRNA was used, please. It is not clear.

A8: Liposomal MMP3-siRNA was used to enhance the stability, specificity, and delivery efficiency of siRNA in ovarian cancer cells. Nanoparticle-based carriers, particularly liposomes, protect siRNA from degradation, facilitate targeted delivery, and reduce toxicity, overcoming key challenges such as rapid degradation and off-target effects.

In the context of cisplatin-resistant ovarian cancer, targeting MMP3 with siRNA offers a promising therapeutic approach. Liposomal formulations ensure that the siRNA reaches the tumor microenvironment effectively, minimizing exposure to healthy cells while maximizing gene silencing in resistant cancer cells. This strategy aims to counteract MMP3-driven chemoresistance, potentially improving treatment outcomes in high-grade serous ovarian carcinoma (HGSOC). We added this to the main text (lines 613-621).

Q9: The conclusion in general is poor, there are several of relevant information described in the paper and the conclusion is very simple, you can improve it.

A9: Thank you for your feedback. We have significantly improved the conclusion, incorporating key findings from the study to provide a more comprehensive summary (lines 860-877).

Q10: The Original Images for Blots/Gels lack a caption or description.

Thank you for your feedback. We have added descriptions for all original blot/gel images.